# Flexible control of representational dynamics in a disinhibition-based model of decision-making

Bo Shen[1]*, Kenway Louie[1,2], Paul Glimcher[1,2]

[1]Neuroscience Institute, New York University Grossman School of Medicine, New York, United States; [2]Center for Neural Science, New York University, New York, United States

**Abstract** Inhibition is crucial for brain function, regulating network activity by balancing excitation and implementing gain control. Recent evidence suggests that beyond simply inhibiting excitatory activity, inhibitory neurons can also shape circuit function through disinhibition. While disinhibitory circuit motifs have been implicated in cognitive processes, including learning, attentional selection, and input gating, the role of disinhibition is largely unexplored in the study of decision-making. Here, we show that disinhibition provides a simple circuit motif for fast, dynamic control of network state and function. This dynamic control allows a disinhibition-based decision model to reproduce both value normalization and winner-take-all dynamics, the two central features of neurobiological decision-making captured in separate existing models with distinct circuit motifs. In addition, the disinhibition model exhibits flexible attractor dynamics consistent with different forms of persistent activity seen in working memory. Fitting the model to empirical data shows it captures well both the neurophysiological dynamics of value coding and psychometric choice behavior. Furthermore, the biological basis of disinhibition provides a simple mechanism for flexible top-down control of the network states, enabling the circuit to capture diverse task-dependent neural dynamics. These results suggest a biologically plausible unifying mechanism for decision-making and emphasize the importance of local disinhibition in neural processing.

*For correspondence:
bs3667@nyu.edu

**Competing interest:** The authors declare that no competing interests exist.

## Editor's evaluation

This novel theoretical work outlines a unifying architecture for decision-making via disinhibition. The model clearly links observations across multiple empirical studies and highlights how characteristics from previous decision models can be effectively integrated into a single mechanism. This will be of interest to a wide variety of neuroscientists who work across levels of analysis.

## Introduction

Inhibition is an essential component in neural network models of decision-making. In standard decision models, pools of option-selective excitatory neurons compete in a winner-take-all (WTA) selection process via feedback inhibition (*Roach et al., 2023*; *Wang, 2002*; *Wong and Wang, 2006*). Generally, such inhibition is thought to be homogeneous and non-selective, with a single pool of inhibitory neurons receiving broad excitation, and in turn inhibiting excitatory neurons. However, more recent empirical findings suggest that inhibitory neurons interact with the decision circuit in a more structured manner. Inhibitory neurons active in decision-making exhibit choice-selective activity on par with excitatory neurons in the frontal cortex (*Allen et al., 2017*), parietal cortex (*Allen et al., 2017*; *Najafi et al., 2020*), and striatum (*Gage et al., 2010*) in contrast to the non-selective or broadly

tuned inhibition seen in visual cortex during stimulus representation (*Bock et al., 2011*; *Chen et al., 2013*; *Hofer et al., 2011*; *Kerlin et al., 2010*; *Liu et al., 2009*; *Niell and Stryker, 2008*; *Sohya et al., 2007*). At an anatomic level, inhibitory interneurons also exhibit a remarkable diversity in morphology, connectivity, and physiological functions (*Kepecs and Fishell, 2014*; *Markram et al., 2004*; *Tremblay et al., 2016*). A prominent circuit motif observed in these anatomical studies is local disinhibition in which vasoactive intestinal peptide (VIP)-expressing interneurons inhibit the neighboring interneurons expressing somatostatin (SST) or parvalbumin (PV) that inhibit dendritic or perisomatic areas in pyramidal neurons, thus locally disinhibiting the activities of the pyramidal neurons in the neighboring area (*Chiu et al., 2013*; *Fino and Yuste, 2011*; *Fu et al., 2014*; *Karnani et al., 2014*; *Karnani et al., 2016*; *Lee et al., 2013*; *Letzkus et al., 2011*; *Pfeffer et al., 2013*; *Pi et al., 2013*; *Urban-Ciecko and Barth, 2016*). Here, we explore the computational implications of that motif in decision-making.

While disinhibitory circuit motifs have been implicated in cognitive processes including learning, attentional selection, and input gating (*Fu et al., 2014*; *Letzkus et al., 2011*; *Wang and Yang, 2018*), how disinhibition functions in decision-making circuits is unknown. Local circuit inputs to the VIP neurons suggest that disinhibition may be a key mechanism for generating the mutual competition necessary for option selection in decision-making. In addition, given the existence of long-range inputs (*Kepecs and Fishell, 2014*; *Lee et al., 2013*; *Pfeffer et al., 2013*; *Pi et al., 2013*; *Schuman et al., 2021*) and neuromodulatory inputs (*Alitto and Dan, 2012*; *Fu et al., 2014*; *Pfeffer et al., 2013*; *Prönneke et al., 2020*; *Rudy et al., 2011*; *Tremblay et al., 2016*) to the VIP neurons, local disinhibition has been proposed to play a particular role in dynamic gating of circuit activity; such gating may be essential in decision circuits underlying flexible behavior, mediating top-down control of network function (*Fu et al., 2014*; *Kamigaki, 2019*; *Lee et al., 2013*; *Letzkus et al., 2011*; *Pi et al., 2013*; *Schuman et al., 2021*; *Zhang et al., 2014*). Here, we hypothesize that disinhibition controls a transition between information processing states, allowing a single decision-making circuit to both represent the values of alternatives and select a single best option amongst those alternatives.

Value representation is prominent in the early stage of a decision. Integrated decision variables combine outcome information such as expected gain and probability of realization. Neural firing rates in numerous decision-related brain areas vary with the integrated option values, including the frontal (*Kiani et al., 2014*; *Kim and Shadlen, 1999*; *Padoa-Schioppa, 2013*; *Padoa-Schioppa and Conen, 2017*; *Pastor-Bernier and Cisek, 2011*; *Roesch and Olson, 2003*; *Thura and Cisek, 2014*; *Thura and Cisek, 2016*; *Yamada et al., 2018*) and parietal (*Andersen and Buneo, 2002*; *Churchland et al., 2008*; *Dorris and Glimcher, 2004*; *Hanks et al., 2014*; *Kiani et al., 2008*; *Kiani et al., 2014*; *Louie and Glimcher, 2010*; *Platt and Glimcher, 1999*; *Roitman and Shadlen, 2002*; *Rorie et al., 2010*; *Shadlen and Newsome, 2001*; *Sugrue et al., 2004*) cortices and basal ganglia (*Ding and Gold, 2010*; *Ding and Gold, 2012*; *Ding and Gold, 2013*; *Thura and Cisek, 2017*). Recent research shows more specifically that neural value coding is contextual in nature, with the value of a given option represented relative to the value of available alternatives (*Churchland et al., 2008*; *Kira et al., 2015*; *Louie et al., 2011*; *Louie et al., 2013*; *Louie et al., 2014*; *Pastor-Bernier and Cisek, 2011*; *Rorie et al., 2010*; *Strait et al., 2014*; *Yamada et al., 2018*). Furthermore, this relative value coding employs a divisive normalization-like representation (*Hunt et al., 2012*; *Louie et al., 2011*; *Louie et al., 2015*; *Yamada et al., 2018*), a canonical computation prevalent in sensory processing and thought to implement efficient coding principles (*Carandini et al., 1999*; *Carandini and Heeger, 1994*; *Carandini and Heeger, 2012*; *Heeger, 1992*; *Heeger, 1993*; *Schwartz and Simoncelli, 2001*; *Silver, 2010*) and temporal adaptation (*Chau et al., 2020*; *Heeger, 1992*; *Louie et al., 2013*; *Louie et al., 2015*; *Steverson et al., 2019*; *Webb et al., 2014*).

Option selection and categorical choice occur when the decision process progresses beyond simple representation. A common and powerful neural mechanism for this categorical choice is WTA competition (*Wickens et al., 2007*; *Wilson, 2007*). WTA dynamics are widely observed in multiple brain regions: the neural firing rate representing the chosen option or action target increases in concert with selection (often reaching an activity threshold at choice), while firing rates representing the other unchosen option are suppressed (*Churchland et al., 2008*; *Gold and Shadlen, 2007*; *Hanes and Schall, 1996*; *Hanks et al., 2014*; *Lo et al., 2015*; *Lo and Wang, 2006*; *Roitman and Shadlen, 2002*; *Rorie et al., 2010*; *Shadlen and Newsome, 2001*; *Wang, 2002*; *Wong and Wang, 2006*). The wide prevalence of WTA dynamics in decision-related neural activities suggests that it is a general feature of biological choice.

Existing models have identified core circuit motifs that produce either normalized value representation or WTA selection (*Figure 1*). For normalized value representation, dynamic circuit-based models emphasize a crucial role for both lateral and feedback inhibition (*Lofaro et al., 2014*; *Louie et al., 2014*). In the dynamic normalization model (DNM), paired excitatory and inhibitory neurons represent each choice option (*Figure 1A*); feedforward excitation delivers value inputs, lateral connectivity mediates contextual interactions, and feedback inhibition drives divisive scaling. This simple differential equation model emphasizes the crucial role of lateral connectivity and feedback inhibition in driving empirically observed divisive scaling and contextual interactions (*Figure 1B*).

For WTA selection, the predominant class of decision models (recurrent network models, hereafter RNM) proposes a central role for recurrent connectivity (*Houck and Person, 2014*; *Ito, 2002*; *Ito, 2006*; *Ito, 2008*; *Llinás, 1975*; *Sathyanesan et al., 2019*; *Sillitoe and Joyner, 2007*) and non-selective feedback inhibition (*Wickens et al., 2007*; *Wilson, 2007*; *Figure 1C*). RNMs capture psychophysical and neurophysiological results in perceptual (*Furman and Wang, 2008*; *Wang, 2002*; *Wong et al., 2007*; *Wong and Wang, 2006*) and economic (*Hunt et al., 2012*; *Jocham et al., 2012*; *Rustichini and Padoa-Schioppa, 2015*; *Soltani and Wang, 2006*) choices, recapitulating much of the complex nonlinear dynamics of empirical neurons (*Figure 1D*). The competitive nature of the RNM generates attractor states which maintain continued activity even in the absence of stimuli, consistent with persistent spiking activity associated with working memory during delay intervals (*Brunel and Wang, 2001*; *Compte et al., 2000*; *Constantinidis et al., 2018*; *Furman and Wang, 2008*; *Hart and Huk, 2020*; *Lo and Wang, 2006*; *Macoveanu et al., 2006*; *Murray et al., 2017*; *Tegnér et al., 2002*; *Wang et al., 2013*; *Wang, 1999*, *Wang, 2002*; *Wong and Wang, 2006*).

While sequential valuation and selection processes may occur independently, electrophysiological evidence shows sequentially coexisting value coding and WTA signals in many prominent decision-related circuits. When decisions are framed as action selection, such integrated representation of values exists primarily in frontoparietal areas tightly linked to motor action commitment. In the control of eye movements, valuation and selection dynamics coexist in multiple brain regions including the lateral intraparietal (LIP) cortex (*Louie and Glimcher, 2010*; *Roitman and Shadlen, 2002*; *Rorie et al., 2010*; *Shadlen and Newsome, 2001*; *Sugrue et al., 2004*), the frontal eye fields (*Ding and Gold, 2012*; *Kim and Shadlen, 1999*; *Roesch and Olson, 2003*), and the superior colliculus (*Basso and Wurtz, 1997*; *Basso and Wurtz, 1998*; *Horwitz et al., 2004*; *Horwitz and Newsome, 1999*; *Zhang et al., 2021*). In these areas, neural activity initially represents the relevant decision variables but shifts to encode the selected saccade after a WTA-like interval. Similar activity emerges in parallel circuits controlling arm movements, including the parietal reach region (*Kubanek et al., 2015*; *Rajalingham et al., 2014*; *Snyder et al., 1997*), dorsal premotor cortex (*Cisek and Kalaska, 2005*; *Pastor-Bernier and Cisek, 2011*; *Thura and Cisek, 2016*), and primary motor cortex (*Thura and Cisek, 2014*). Notably, when examined, contextual value coding during a decision typically arises after the initial absolute value coding (*Louie et al., 2014*; *Pastor-Bernier and Cisek, 2011*; *Rorie et al., 2010*), consistent with a local normalization process; these dynamics suggest that normalized value coding is not simply inherited from upstream regions and support coexisting within-region normalization and selection computations.

Despite electrophysiological evidence for sequentially coexisting relative value coding and WTA signals in prominent decision-related circuits, no current model integrates both properties within a single circuit. The DNM cannot capture late-stage choice dynamics because it lacks a mechanism for WTA competition. Similarly, RNMs typically neither exhibit contextual value coding nor predict contextual choice patterns (*Wang, 2012*) due to the lack of structured lateral inhibition. Here, we propose that disinhibition is a biologically plausible solution to unify these key features of decision-making into a single circuit. We develop and characterize a biological circuit consisting of three neuronal types which critically include a form of local disinhibition. This model hybridizes the architectural features of divisive gain control and recurrent self-excitation used in existing models but utilizes disinhibition rather than the commonly assumed pooled inhibition to implement competition. We find that the disinhibition-based model unifies multiple characteristics of decision activity including normalized value coding, WTA choice, and working memory. A top-down gating signal operating via this disinhibition enables the model to switch between the states of value representation and WTA selection and to reproduce decision activity in a range of experimental paradigms with diverse task timing and activity dynamics. These findings suggest that local disinhibition

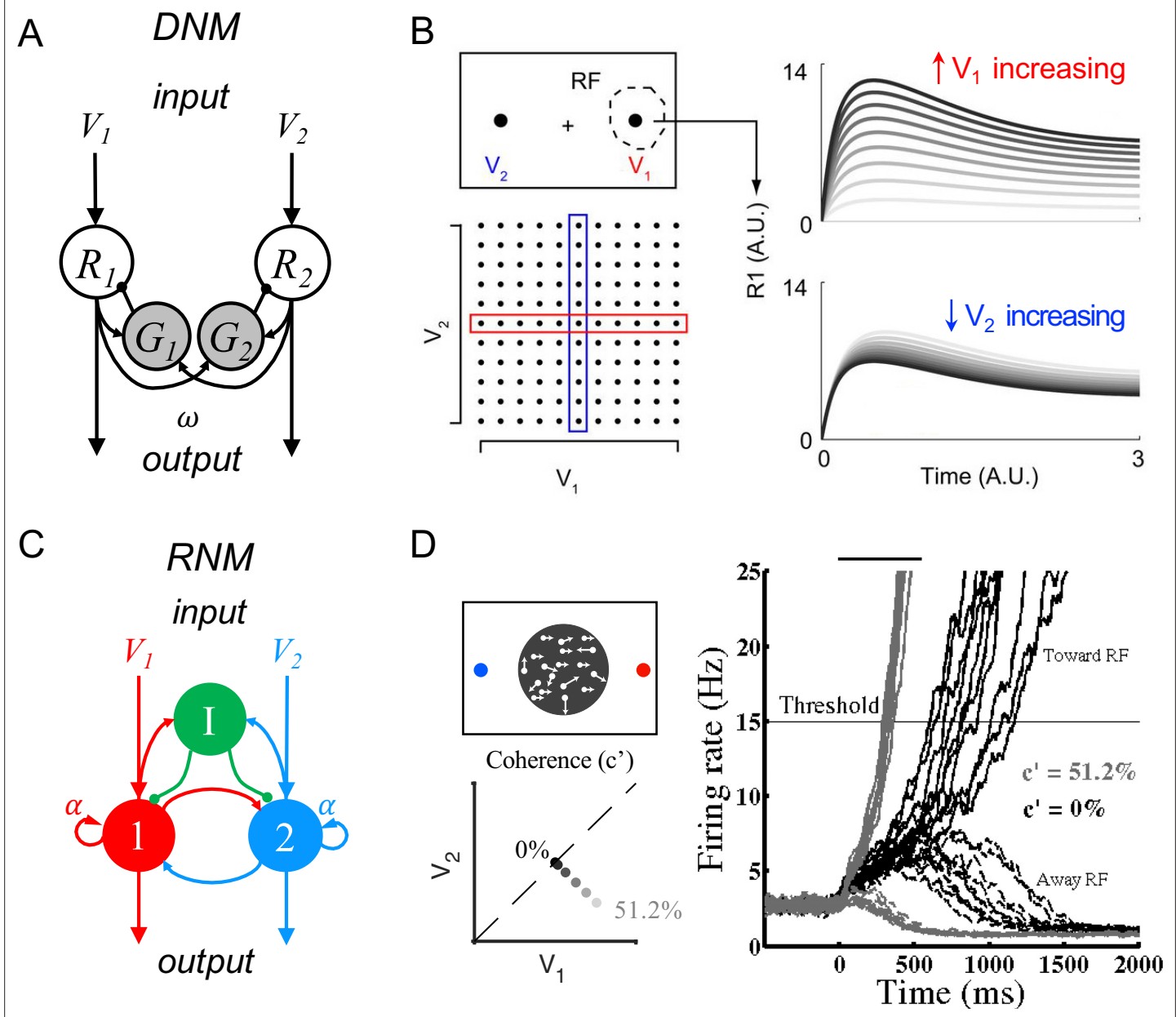

**Figure 1.** Standard circuit motifs and neural dynamics in existing decision-making models. (**A**) Dynamic normalization model (DNM). Each pair of excitatory ($R$) and inhibitory ($G$) units corresponds to an option in the choice set, with $R$ receiving value-dependent input $V$ and providing output. Lateral interactions implement a cross-option gain control that produces normalized value coding. Panel adapted from Figure 1 from *Louie et al., 2014* (**B**) DNM predicted dynamics replicate empirical contextual value coding. The example task involves orthogonal manipulation of both option values. $R_1$ activity increases with the direct input value $V_1$ (array framed in red) but is suppressed by the contextual input $V_2$ (array framed in blue), consistent with value normalization. (**C**) Recurrent network model (RNM). The network consists of excitatory pools with self-excitation (1 and 2) and a common pool of inhibitory neurons (*I*). Panel adapted from *Wong and Wang, 2006*. (**D**) RNM predicted dynamics generate winner-take-all selection. The example task involves motion discrimination of the main direction of a random dot motion stimulus with varying coherence (*c'*) levels (left). Model activity (right) under two different levels of input coherence (0 and 51.2%) predicts different ramping speeds to the decision threshold and generates a selection even with equal inputs.

provides a robust, biologically plausible integration of normalization and WTA selection in a single-circuit architecture.

## Results

### Local disinhibition decision model

To develop an integrated circuit model of decision-making, we systematically tested a series of models incorporating disinhibitory motifs and the core elements of existing models, namely divisive gain control, recurrent excitation, and mutual competition (*Figure 2—figure supplement 1*; see **Methods** *Motifs tested and compared for normalized coding and WTA choice* for the analysis details). This analysis identified *local disinhibition* as the crucial component that can integrate mutual competition and value normalization within the existing circuit architecture of DNM. In the rest of this paper, outside of the methods and supplementary figures, we focus on this local disinhibition decision model (hereafter LDDM) that emerged from our detailed examination of potential models.

In the LDDM (*Figure 2A*), as in the DNM, option-specific excitatory *R* units receive value inputs and interact via widespread lateral inhibition. However, the LDDM also includes an option-specific disinhibitory *D* unit that receives input from its associated excitatory *R* unit and locally inhibits the inhibitory *G* unit in the local circuit. In this way, disinhibition biased by different value inputs can serve to selectively release local circuit gain control, generating an unbalanced gain control between local and opponent circuits and leading to a WTA competition. In this model, the network thus shifts from value coding to WTA competition regimes in response to the onset of disinhibition (controlled by the coupling strength between *R* and *D*). With zero or weak *R-D* coupling, the circuit preserves normalized value coding consistent with the DNM; with strong *R-D* coupling, the circuit switches to a state of WTA selection (*Figure 2B*). Inhibitory units, as a result, dynamically switch from a non-selective response pattern to a selective response pattern (*G* and *D* units in *Figure 2B*) driven by local disinhibition. This flexible onset of disinhibition is modeled after biological findings, which show that activation of disinhibition in cortical circuits arises from exogenous, long-distance projections (*Fu et al., 2014*; *Kamigaki, 2019*; *Lee et al., 2013*; *Pi et al., 2013*; *Zhang et al., 2014*; *Figure 2C*). This form of top-down control allows for flexibility in the relative timing of the valuation and selection processes, consistent with neural and behavioral data in different task paradigms (see *Gated disinhibition provides top-down control of choice dynamics*).

Activity dynamics of the LDDM are described by a set of differential equations:

$$\tau_R \frac{dR_i}{dt} = -R_i + \frac{V_i + \alpha R_i + B_R}{1 + G_i}, \tag{1}$$

$$\tau_G \frac{dG_i}{dt} = -G_i + \sum_{j=1}^{N} \omega_{ij} R_j + B_G - D_i, \tag{2}$$

$$\tau_D \frac{dD_i}{dt} = -D_i + \beta R_i. \tag{3}$$

where *i*=1, …, *N* designates choice alternatives, each of which is represented by an *R* unit receiving selective input $V_i$ and non-selective baseline input $B_R$. $\tau_R$, $\tau_G$, and $\tau_D$ are the time constants for the *R*, *G*, and *D* units. The weights $\omega_{ij}$ represent the coupling strength between excitatory units $R_j$ and inhibitory (gain control) units $G_i$, with each *G* unit driven by a weighted sum of excitatory inputs from all *R* units and a non-selective baseline input $B_G$ and inhibited by its local $D_i$; the parameter $\alpha$ reflects the strength of recurrent self-excitation on *R* units. Finally, $\beta$ weights the coupling strength between the excitatory $R_i$ and the disinhibitory $D_i$ units and is presumed to be under external (task-triggered) control.

### Dynamic divisive normalization preserved in the LDDM

We first examine whether the LDDM retains the dynamics of divisively normalized value coding seen empirically and in the DNM (*Lofaro et al., 2014*; *Louie et al., 2014*). As discussed above, during the initial option evaluation, the disinhibitory units are silent ($\beta = 0$); therefore, the sole difference between the LDDM and the DNM is recurrent excitation (controlled by $\alpha$). Example activity traces in *Figure 3B* show that the LDDM preserves characteristic early-stage dynamics and contextual modulation seen in both empirical data (*Figure 3C*) and the original DNM (*Lofaro et al., 2014*; *Louie et al., 2011*; *Louie et al., 2014*). Immediately after stimulus onset, $R_1$ activities replicate the transient peak observed in a wealth of studies (*Andersen and Buneo, 2002*; *Churchland et al., 2008*; *Gnadt and*

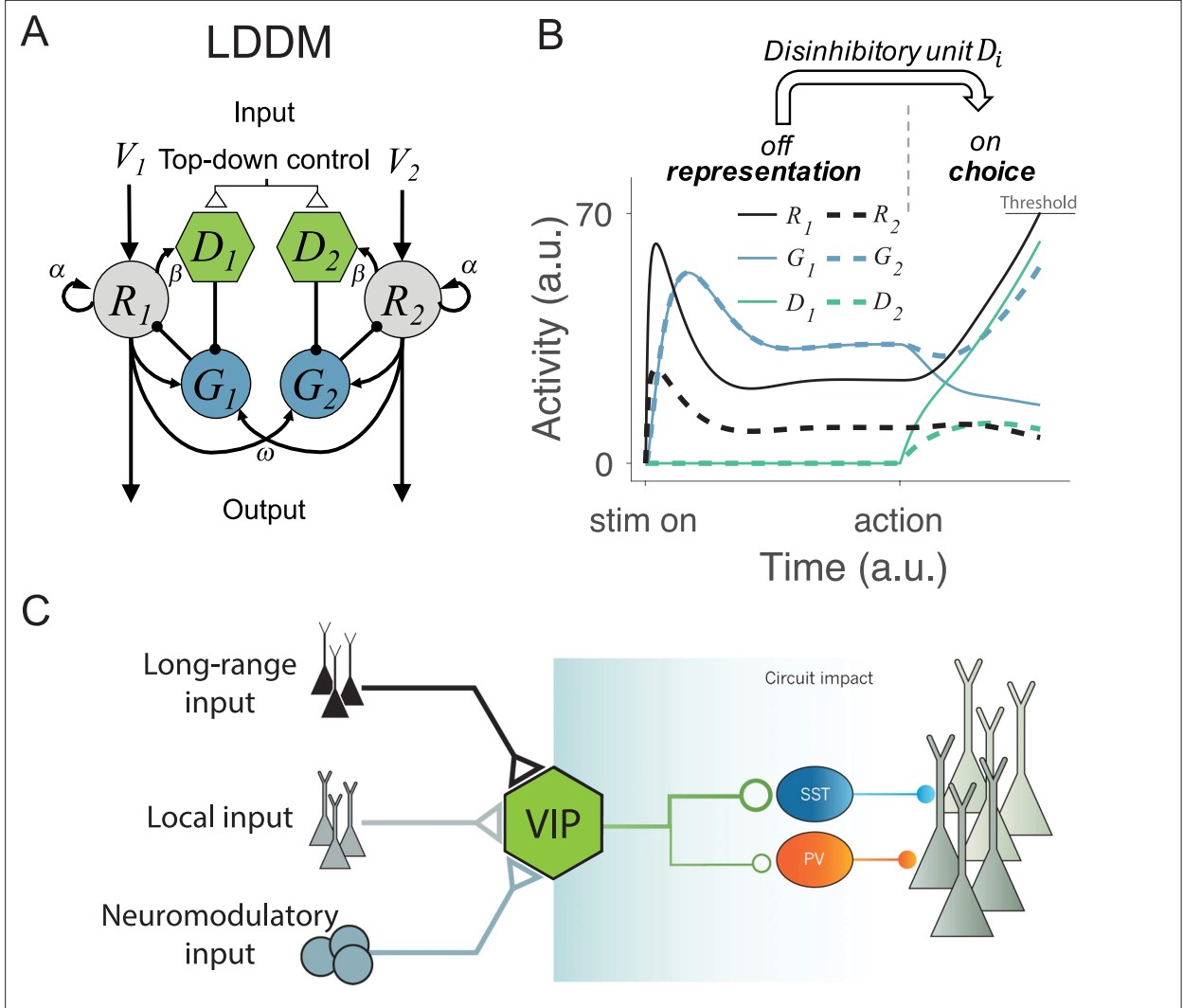

**Figure 2.** Local disinhibition decision model (LDDM) and its biological plausibility. (**A**) LDDM extends the dynamic normalization model (DNM) by incorporating a disinhibitory *D* unit to mediate the local disinhibition of the associated excitatory *R* unit; strength of *R* to *D* coupling is controlled by the parameter *β* presumed via an external top-down control. $V_i$, *α*, and *ω* indicate the corresponding input value to each option, self-excitation of *R* unit, and the coupling weights from *R* to *G* unit, respectively. (**B**) The network phase transition between representation and choice under gated disinhibition. With the disinhibitory module silent, the network performs dynamic divisive normalization on *R* units and predicts non-selective inhibition via *G* units; after the disinhibitory module is triggered via an external top-down control signal, the network switches to a winner-take-all competition dynamic. The circuit predicts selective inhibition after disinhibition is triggered. (**C**) Biological basis of disinhibition. Disinhibition provides a mechanism for dynamic gating of circuit states. Vasoactive intestinal peptide (VIP)-expressing interneurons typically inhibit somatostatin (SST) and parvalbumin (PV)-positive interneurons, resulting in a disinhibition of pyramidal neurons. VIP neurons receive local, long-range, and neuromodulatory input, providing different potential mechanisms to modulate local circuit dynamics.

© 2014, Springer Nature. Panel C is reproduced from Figure 3 from *Kepecs and Fishell, 2014* with permission from Springer Nature. It is not covered by the CC-BY 4.0 licence and further reproduction of this panel would need permission from the copyright holder.

The online version of this article includes the following figure supplement(s) for figure 2:

**Figure supplement 1.** Testing and comparing different dynamic normalization model (DNM) modifications for integrating normalized value coding and winner-take-all (WTA) competition.

*Andersen, 1988*; *Louie et al., 2011*; *Louie et al., 2014*; *Platt and Glimcher, 1999*; *Rorie et al., 2010*; *Sugrue et al., 2004*). Furthermore, the network settles to equilibrium displaying *relative* value coding: $R_1$ activity increases with $V_1$ and decreases with $V_2$, reflecting a contextual representation of value (*Figure 3B*, $R_1$ activity across $V_1$ inputs [upper panel] and $V_2$ inputs [bottom panel]).

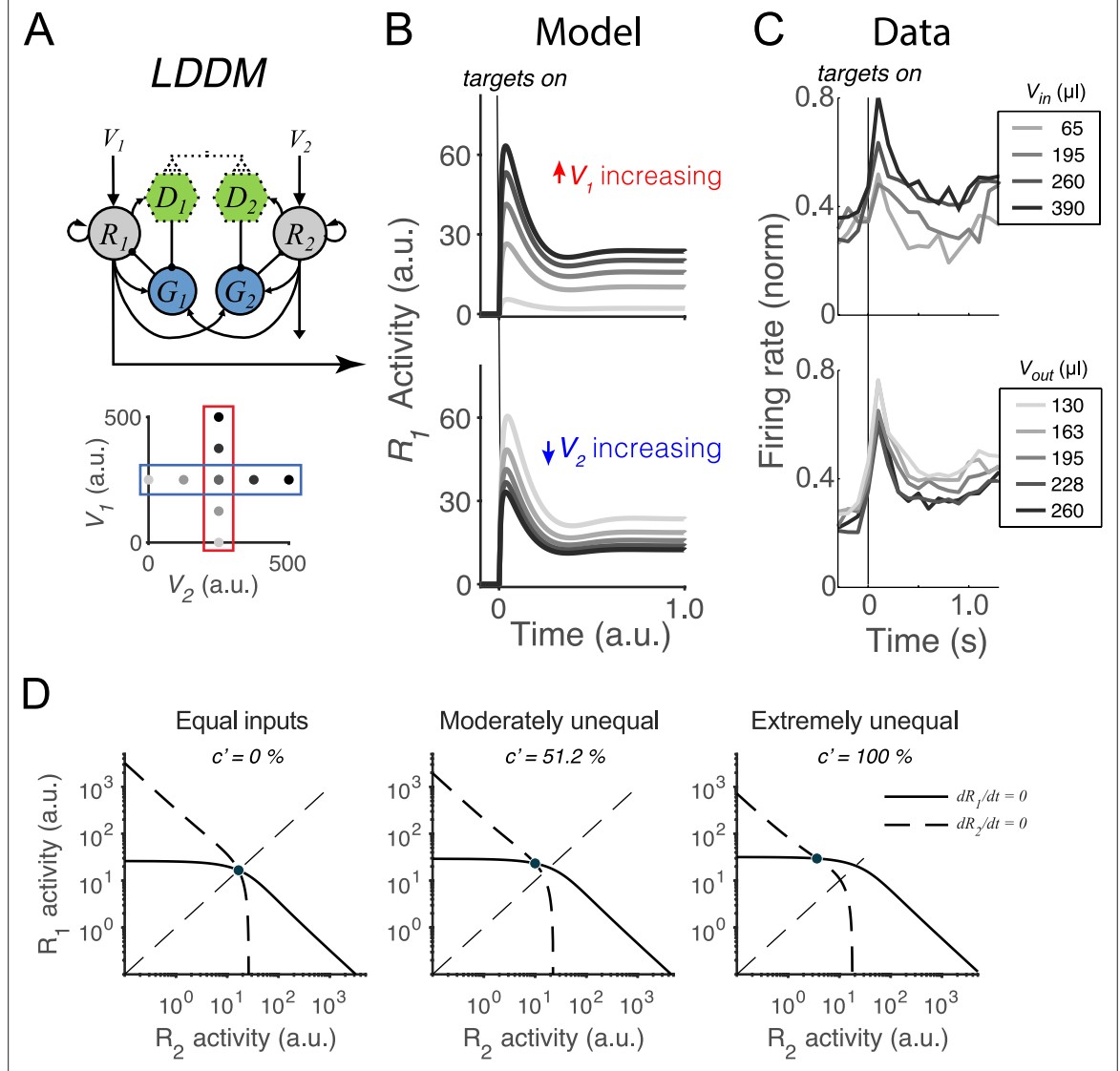

**Figure 3.** Normalized value coding in the local disinhibition decision model (LDDM). (**A**) In this example, the LDDM receives a set of two input values with varying $V_1$ (framed in red) and $V_2$ (framed in blue). (**B**) Example LDDM dynamics show relative value coding. $R_1$ activity shows a transient peak before a sustained period of coding. Increasing $V_1$ increases $R_1$ activity but increasing $V_2$ decreases $R_1$ activity. (**C**) Value coding dynamics recorded in monkey parietal cortex. The model prediction we showed is consistent with the empirical observation. Panel is adapted from Figure 1B and D from *Louie et al., 2011*. (**D**) Phase plane analysis of the system under equal (left), weakly unequal (middle), and extremely unequal (right) inputs. The nullclines of $R_1$ (solid) and $R_2$ (dashed) indicating the equilibrium state of the individual units intersect at a unique and stable equilibrium point with divisively normalized coding.

Taking advantage of its simplified mathematical form, we analytically evaluated the LDDM by conducting phase plane analyses. We found that it represents each set of input values $(V_1, \ldots, V_N)$ as one unique and stable equilibrium point in its output space $(R_1, \ldots, R_N)$ when $\beta = 0$. Specifically, we solved for the equilibrium state of each $R$ unit by setting each differential equation (*Equations 1–3*) to zero, which defines the nullcline of each $R$ unit as a function of the activity of the complementary $R$ unit, visualized in *Figure 3D*. The nullclines of $R_1$ (solid) and $R_2$ (dashed) intersect at a unique equilibrium point, regardless of whether input values are equal or unequal (see different panels for examples of different inputs). This point indicates that the dynamical system, when receiving any positive inputs, can maintain a unique equilibrium where every unit maintains a steady level of activity. Linearization analysis around this point suggests that this point is attractive: given any initial values to the system, the activities of the units will converge into the unique equilibrium point for the network (see **Methods** *Equilibria and stability analysis of the LDDM* for mathematical proof). The steady state of

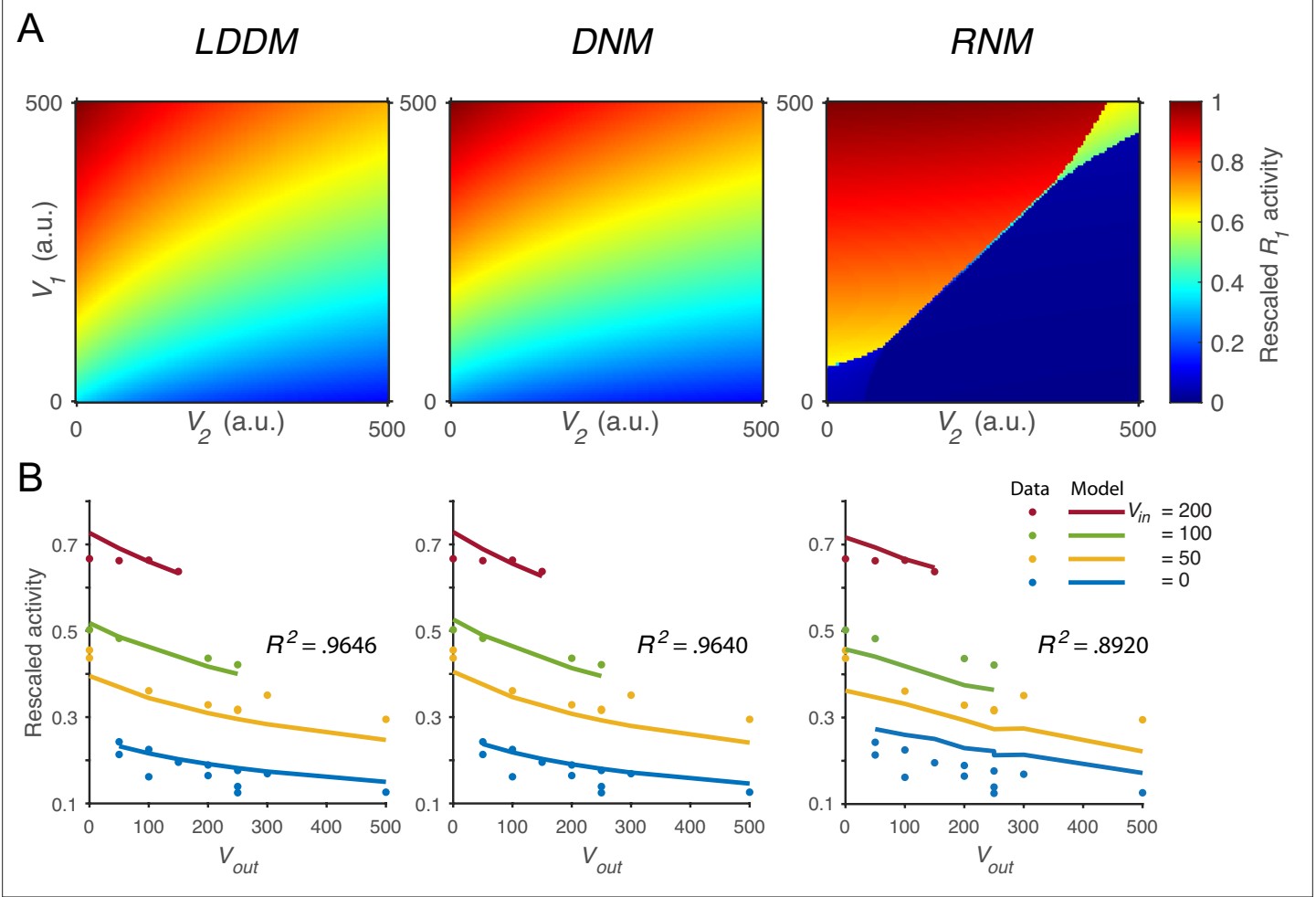

**Figure 4.** Quantitative comparison of contextual value coding across the local disinhibition decision model (LDDM), dynamic normalization model (DNM), and recurrent network model (RNM) models. (**A**) Comparison between the LDDM (left), the DNM (middle), and the RNM (right) in value coding. The LDDM and the DNM show normalized value coding. The neural activity of $R_1$ (indicated by color) increases with the direct input $V_1$ but decreases with the contextual input $V_2$. The LDDM shows slightly stronger contextual modulation than the DNM but qualitatively replicated normalized value coding. The RNM shows a qualitatively different pattern consistent with winner-take-all (WTA) competition. Within the regime of WTA competition ($V_1$ and $V_2$ within a reasonable scale), $R_1$ activity is high when $V_1 > V_2$ and low when $V_1 < V_2$. (**B**) Fitting the models to a trinary choice dataset shows that the LDDM (left panel) performed slightly better than the DNM (middle panel) in capturing the neural activities responding to values inside ($V_{in}$) and outside ($V_{out}$) of the receptive field. Fitting the RNM to the dataset does not capture the neural activities as well as the LDDM (and DNM; right panel).

The online version of this article includes the following figure supplement(s) for figure 4:

**Figure supplement 1.** Parameter recovery in fitting the local disinhibition decision model (LDDM) to normalized value coding data.

**Figure supplement 2.** Recurrent network model (RNM) activity fit to normalized value coding data.

neural activity at equilibrium (noted as $R_i^*$) reflects divisive normalization (*Equation 4*), as in the original DNM (*Lofaro et al., 2014*; *Louie et al., 2014*). The only difference between the LDDM and the DNM at equilibrium is the introduction of a constant in the denominator ($B_G - \alpha$) representing baseline gain control and recurrent excitation; this change rescales the activity magnitudes but preserves normalized value coding.

$$R_i^* = \frac{V_i + B_R}{1 + B_G - \alpha + \sum_{j=1}^{N} \omega_{ij} R_j^*} \qquad (4)$$

We next verified that the normalized value coding produced by the LDDM cannot be implemented by standard RNM models. *Figure 4A* compares the activity of $R_1^*$ as a function of both value inputs ($V_1$ and $V_2$) in the LDDM (left panel), the original DNM (middle panel), and the RNM (right panel). Both the

LDDM and the DNM exhibit $R_1^*$ activities (indicated by color) that monotonically increase with input $V_1$ but decrease with $V_2$, with a slightly steeper $V_2$ dependence in the LDDM versus the DNM model depending on the rescaling of $\alpha$. In contrast, strong WTA dynamics in the RNM implement categorical (choice) coding rather than relative value representation, with high or low coding of input values (right panel).

To quantitatively test value normalization, we fit the models to observed firing rates of monkey LIP neurons under varying reward conditions (*Louie et al., 2011*). In the empirical data (*Figure 4B*, dots), LIP activity increases with the reward (water quantity) associated with the target inside the neuronal response field ($V_{in}$) and decreases with the summed rewards of targets outside the response field ($V_{out}$). The fitting results show that the DNM captures the rescaled firing rates very well with only two free parameters (baseline input $B_R$ = 70.92 and an arbitrary scaling parameter $R_{max}$; see **Methods**; middle panel in *Figure 4B*, $R^2$=0.9640). The LDDM with an additional parameter $B_G - \alpha$ introduced by self-excitation and baseline gain control fitted slightly better than the DNM ($B_R$ =71.53, $B_G - \alpha = 3.82$; see **Methods**; left panel in *Figure 4B*, $R^2$=0.9646; parameter recovery analysis shows that the LDDM is highly robust in the data fitting, *Figure 4—figure supplement 1*). Note that fitting to the current dataset is not able to differentiate the contributions of $\alpha$ and $B_G$ to the neural dynamics (see proof in **Methods**); thus more empirical data will be needed to draw conclusions about the role of recurrent self-excitation in value coding. However, we do show below that self-excitation is critical for generating persistent activities (see section: *Disinhibition controls point versus line attractor dynamics in persistent activity*).

We found that fitting the standard RNM with its standard four parameters (see **Methods**) cannot capture the pattern of neural activity as well as the LDDM and DNM (right panel in *Figure 4B*; $R^2$=0.8920). This small but clear difference in performance between model classes arises from the difference between divisive (DNM and LDDM) and subtractive (RNM) types of inhibition, with subtractive inhibition failing to capture the concave contextual effects predicted by divisive models. Furthermore, fitting the RNM to the data results in a parameter regime that can no longer generate WTA competition; instead, the model predicts mean firing rates in a low-activity regime with a maximum value of 3.5 Hz (*Figure 4—figure supplement 2*). These results suggest that RNM models cannot simultaneously support both normalized value coding and WTA selection regimes.

## Local disinhibition drives WTA competition

A key question is whether the LDDM can also produce WTA competition. Given the architecture of the LDDM, local disinhibition is hypothesized to break the symmetry between option-specific *R-G* sub-circuits, enabling a competitive interaction between sub-circuits. To examine whether this competition produces WTA selection, we simulated model activity in a reaction-time version of a motion discrimination task, a standard perceptual decision-making paradigm in non-human primates (*Churchland et al., 2008*; *Roitman and Shadlen, 2002*). The task contains two stages of processing: the pre-motion stage with only the choice targets presented and the motion stage presenting a random-dot motion stimulus simultaneously with a go signal. Animals are allowed to select an option, indicating their perception of the main direction of the motion, at any time following motion stimulus/go signal onset (see timeline, *Figure 5A*). During the pre-motion stage, we simulated equal value inputs, given the equal prior probability of either target being correct in the standard task. The simulated dynamics replicate the characteristic transient peak observed in both perceptual and economic decision-making tasks (*Andersen and Buneo, 2002*; *Churchland et al., 2008*; *Louie et al., 2011*; *Rorie et al., 2010*). At motion stimulus onset, inputs to the two *R* units are changed according to the task design; disinhibition (i.e. $\beta$ value) is switched on at the go signal, simultaneously with motion inputs.

We find that the LDDM replicates neural and behavioral aspects of WTA competition. In *Figure 5A*, we show example model activities for five input strengths corresponding to different motion coherence levels. Consistent with electrophysiological recordings in the posterior parietal cortex (*Churchland et al., 2008*; *Roitman and Shadlen, 2002*; *Shadlen and Newsome, 2001*), model *R* unit activities bifurcate based on the input strengths, with the unit receiving stronger input ramping-up to an (arbitrary) decision threshold while the activity of the opponent unit is suppressed. The speed of bifurcation depends on the contrast between the inputs, a variable equivalent to motion coherence in the experimental literature (*Roitman and Shadlen, 2002*; *Shadlen and Newsome, 2001*). Furthermore, the LDDM predicts the dynamics of the two types of interneurons *G* and *D* governing excitatory neuron

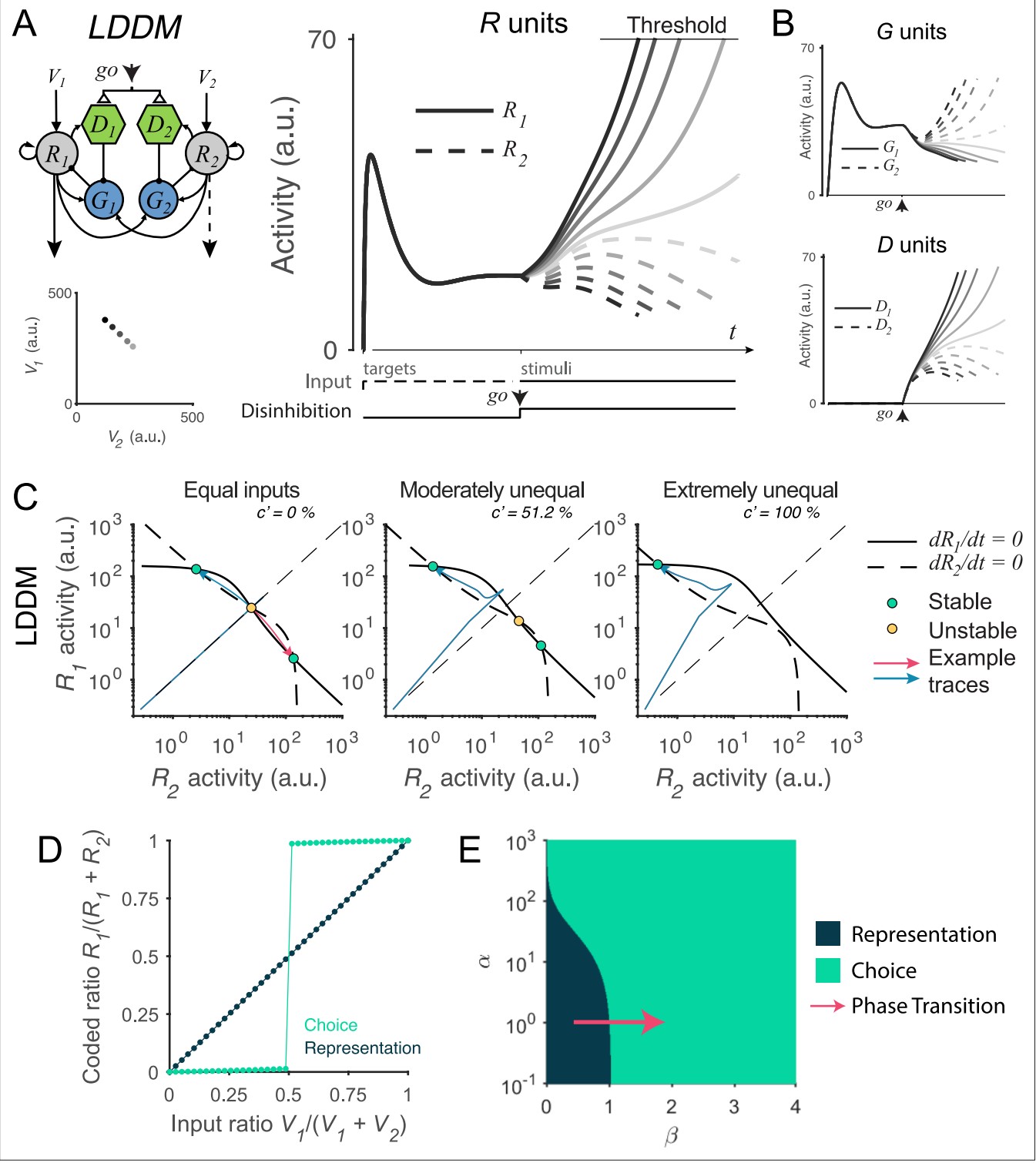

**Figure 5.** Recurrent network model (RNM)-like winner-take-all (WTA) selection dynamics in the local disinhibition decision model (LDDM). (**A**) Example $R_1$ (solid) and $R_2$ (dashed) dynamics in a classic reaction-time motion discrimination task. The model predicts phasic stimulus onset dynamics during the pre-stimulus stage and WTA dynamics during the stimulus stage when receiving different input values (left inset). Consistent with RNM dynamics (upper right inset), the $R$ unit receiving stronger input ramps up to reach the decision threshold while the opponent $R$ unit activity is suppressed; the speed of bifurcation depends on the input strength. (**B**) The model predicted the dynamics of $G$ units (top) and $D$ units (bottom). (**C**) Phase plane analysis of the LDDM (lower) compared with the original RNM (upper inset) shows the basis for WTA dynamics under equal (left), moderately unequal (middle), and extremely unequal (right) inputs. Both models show similar features across input values: under equal inputs, the nullclines of $R_1$ and $R_2$ intersect on

*Figure 5 continued on next page*

*Figure 5 continued*

three equilibrium points, with one unstable point (yellow) and two stable attractors (green; left). Under unequal inputs, the basin of nullclines is biased to the side with stronger input (middle). When the inputs are strongly biased, only the attractor associated with stronger input retains (right). Red and blue lines show example traces of $R_1$ and $R_2$ activities. (**D**) A comparison of the coded ratio between the representation (black) and WTA competition (green) regimes. While the LDDM preserves the input ratios during value representation, it shifts to a categorical coding of choice during WTA selection. (**E**) Distinct normalized value coding (dark) and WTA competition (green) regimes in the parameter space defined by $\alpha$ and $\beta$. Across a wide range of $\alpha$, the transition between valuation and selection regimes can be implemented by an increase in $\beta$ (pointed by the red arrow).

The online version of this article includes the following figure supplement(s) for figure 5:

**Figure supplement 1.** Phase plane analyses of the local disinhibition decision model (LDDM) across a wide range of recurrent excitation strengths ($\alpha$) and local disinhibition strengths ($\beta$).

computation (**Figure 5B**). Prior to the go signal, the two *G* units share the same activity. However, after the go signal, the activity levels bifurcate because of disinhibition. In contrast to *R* units, the *G* unit in the sub-circuit receiving stronger input shows lower activity, indicating a stronger disinhibition of the associated *R* unit. Thus, the LDDM exhibits mutual competition that generates WTA selection in excitatory neurons, as in the existing RNM; this competition is mediated by a novel disinhibitory control input achieved through the use of biologically identified different interneuron subtypes.

What features of the LDDM are essential to generate WTA competition? We examined the dynamical properties of the system under disinhibition by conducting phase plane analyses. As shown in **Figure 5C**, the network in the choice regime ($\beta_{on} = .9$ in this example) shows a different configuration of nullcline intersections than the network in the value representation regime ($\beta_{off} = 0$; **Figure 3D**). Given equal (value) inputs, the nullclines of $R_1$ and $R_2$ intersect at three equilibrium points (left panel in **Figure 5C**), with the central point unstable and the two peripheral points stable. Thus, given an initial configuration of $R_1$–$R_2$ activities (with the presence of noise), the system will converge to the closer peripheral attractor (see example activity traces in blue and red thin lines) and implement WTA competition. Given moderately unequal inputs, the basin of attraction is biased toward the side with higher input, resulting in a higher probability of falling into the side with higher input (middle panel in **Figure 5C**). When inputs are extremely unequal, the stable equilibrium in the middle of the basin and the unstable equilibrium point associated with weaker input no longer exist, leaving only the attractor associated with stronger input (**Figure 5C**, right). Thus, across varying degrees of input coherences, disinhibition drives the LDDM toward selecting one of the potential choices. This can be seen in **Figure 5D** by viewing the output ratio ($\frac{R_1^*}{R_1^*+R_2^*}$) of the preferred attractor as a function of input ratio ($\frac{V_1}{V_1+V_2}$): under active disinhibition ($\beta = .9$) we observe categorical coding (green line), in contrast to under inactive disinhibition ($\beta = 0$) where the output ratio faithfully preserves the original ratio of inputs (dark line; other parameters used in the simulation: $\alpha = 15$, $B_R = 0$, $B_G = 0$, $V_1 = 250 * (1 + c')$ , $V_2 = 250 * (1 - c')$ , and $\omega_{ij} = 1$).

To understand the operating regimes of the LDDM, we quantified model behavior across the full parameter space defined by recurrent excitation weight ($\alpha$) and local disinhibition weight ($\beta$), both of which are critical in determining the properties of the system (see **Methods** *Equilibria and stability analysis of the LDDM* for mathematical proof). Decisions with equivalent inputs are a critical test of WTA behavior since WTA systems should select an option (stochastically) even in these symmetric scenarios (**Furman and Wang, 2008**; **Lo and Wang, 2006**; **Wang, 2002**; **Wong and Wang, 2006**); we therefore analyzed system behavior under equal value inputs. As shown in **Figure 5E**, this analysis revealed two distinct territories corresponding to value representation and WTA-operating regimes. The value representation regime generates a unique attractor for normalized value representation but no WTA attractors; in contrast, the WTA regime (induced by a change in $\beta$) generates no normalization attractor, but instead, $R_1$ and $R_2$ always diverge into high-contrast attractors (see **Figure 5—figure supplement 1** and **Methods** *Equilibria and stability analysis of the LDDM* for a full description of regime parcellation). The $\beta$ value in the WTA regime is always larger than zero even when it asymptotically approaches zero when recurrent excitation is extremely strong, suggesting that disinhibition is always required to generate WTA choices. Models with a wide range of recurrent excitation can shift from value representation to WTA choice with an increase in local disinhibition strength (e.g. red arrow in **Figure 5E**) or under more limited conditions with an increase in recurrent activation. These findings emphasize the impact of changes in local disinhibition to WTA choice and highlight a particular role for a dynamic gating signal in controlling the transition from value coding to option selection.

## The LDDM captures empirical choice behavior and neural activity

While the preceding analyses show that the LDDM can generate value normalization and WTA selection, a critical question is whether this circuit architecture accurately captures empirically observed behavioral and neural aspects of decision-making. Here, we take advantage of the limited number of parameters in this differential equation-based LDDM (compared to more complicated conductance-based biophysical models; *Tegnér et al., 2002*; *Wang, 1999*; *Wang, 2002*; *Wong and Wang, 2006*), which allows model fitting to empirical data. Specifically, we fit LDDM parameters to nonhuman primate behavior from the reaction-time version of the motion discrimination task described above. The choice and reaction time (RT) data from monkeys align with a reduced form model of decision-making (the drift-diffusion model; *Ratcliff and McKoon, 2008*), and the activity of posterior parietal neurons recorded during this task display characteristic decision-related features (motion-dependent ramping, a common decision threshold, and WTA activity).

To fit the LDDM to behaviorally observed RTs, we employed the standard quantile maximum likelihood estimation (QMLE) method to the RT distributions across input coherence levels (0–51.2%), with correct and error trials dissociated (*Hawkins et al., 2015*; *Heathcote et al., 2002*; *Ratcliff and Tuerlinckx, 2002*). We set $\omega_{ij}$ as 1 and the baseline input $B_R$ as zero. Baseline gain control ($B_G$) and self-excitation ($\alpha$) are collinear as mentioned above (see model fitting in *Figure 4*), and this is also true in fitting WTA choice behavior (see *Figure 6—figure supplement 3*). To address this, we kept $\alpha$ as a free parameter but set $B_G$ to zero (note that this limits the interpretability of fit $\alpha$ values as simply the level of recurrence, a point we address below and in the supplementary materials). The model is then reduced to seven parameters: recurrent excitation weight $\alpha$, local disinhibition weight $\beta$, noise parameter $\sigma$, input value scaling parameter $S$, and time constants $\tau_R$, $\tau_G$, and $\tau_D$ (see **Methods** for model-fitting details). Predictions of the best fitting model are shown in *Figure 6A* (best fitting parameters: $\alpha = 0$, $\beta = 1.434$, $\sigma = 25.36$, $S = 3251$, $\tau_R = .1853$, $\tau_G = .2244$, and $\tau_D = .3231$). The optimization surfaces visualized across pairs of parameters (*Figure 6—figure supplement 1*) were consistent with the robust parameter fitting. A parameter recovery analysis indicated that the parameters are recoverable and identifiable within the network (*Figure 6—figure supplement 2*). While there is a small amount of collinearity between $\alpha$ and $\beta$ in the fit to behavioral choice data, further simulation uncovered that these two parameters have notably different effects on the shapes of LDDM-predicted RT distributions: increasing $\beta$ decreases the skewness of the RT distribution; whereas, increasing $\alpha$ increases the skewness. These effects on choice dynamics likely play a role in the ability of the LDDM to perform alternative models in fitting behavioral data (see below) and reinforce the separability and influence of disinhibition and recurrence on behavior (*Palminteri et al., 2017*).

Model-predicted RT distributions (lines) closely follow the empirical distributions (bars) for both correct (blue) and error (red) trials across different levels of input coherence. The aggregated mean choice accuracy and RT data are shown in *Figure 6C*. Model choice accuracy (line) captures the average empirical psychometric function (crosses); model RT captures coherence-dependent changes in the chronometric function, including longer RTs in error trials (dashed line and empty dots) compared to correct trials (solid line and dots). Beyond mean RT data, the LDDM accurately captured aspects of the empirical RT distributions, as evident in the quantile probability plot of RT quantiles as functions of chosen ratio (*Figure 6B*). Given the mathematical collinearity issue between $B_G$ and $\alpha$, it is important to note that the fitted value of $\alpha$ should not be interpreted as reflecting the exact level of recurrence in the circuit. Future empirical data will be needed to differentiate how recurrence and baseline inhibition contribute to the LDDM WTA selection.

We compared the performance of the LDDM in fitting this classical dataset with the reduced form of the RNM (*Wong and Wang, 2006*; *Figure 6—figure supplement 4*), as well as another prominent computational decision model with a similar architecture of mutual inhibition – the leaky competing accumulator (LCA) model (*Usher and McClelland, 2001*; see *Figure 6—figure supplement 5*). The performances of the three models were close in predicting averaged RTs and choice accuracy (panel **C**). However, the LDDM captures the skewness and the shape of RT distributions better than the other two, as reflected in goodness of fit (negative log-likelihood) and Akaike information criterion (AIC) measures ($nLL_{LDDM} = 16,546$, $nLL_{RNM} = 16,573$, $nLL_{LCA} = 16,948$, $AIC_{LDDM} = 33,109$, $AIC_{RNM} = 33,165$, and $AIC_{LCA} = 33,932$).

Notably, the LDDM – fit only to behavior – generates predictions about the underlying neural dynamics that can be compared to electrophysiological findings. We examined $R$ unit activity in the

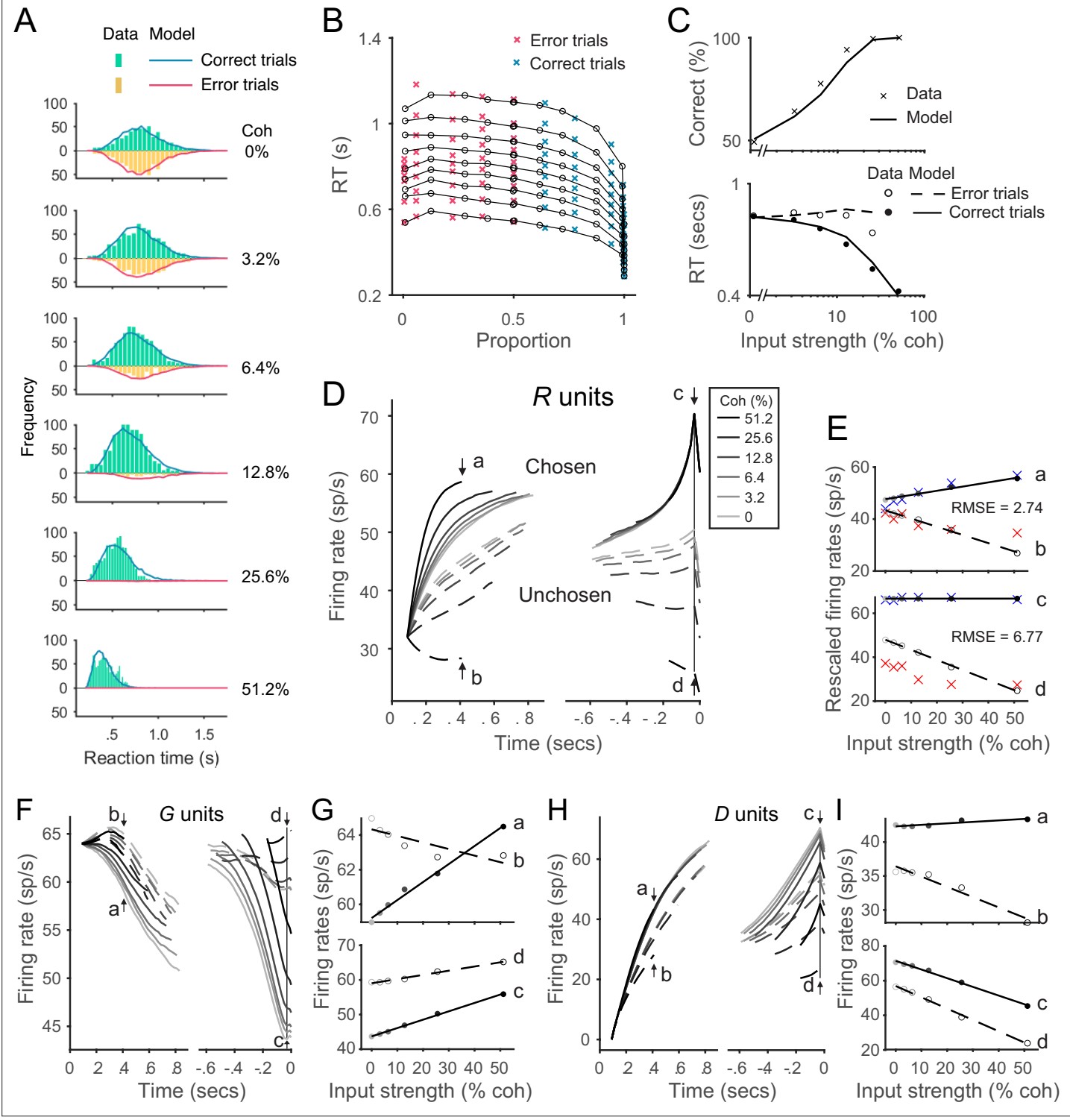

**Figure 6.** The local disinhibition decision model (LDDM) performs well in capturing empirical behavior and neurophysiological data during perceptual decision-making. (**A**) Model predicted RT distributions fit to behavioral data. Predicted RT distribution (lines) match the histogram of empirical RT distribution (bars), with correct and error trials separately (indicated by color) across levels of input strength (% coherence). (**B**) The best-fit model predictions of the LDDM and the original recurrent network model (RNM; upper-right inset) are visualized in quantile probabilities. Nine quantiles of RT under each condition are stacked on the x-axis indicating the correct choice proportion under each input coherence (0–0.5 are error trials, shown in red crosses; .5–1 are correct trials, shown in green crosses). LDDM predicts the choice proportion and the shape of RT distribution as well as the original RNM. (**C**) Model predicted psychometric function (upper) and chronometric function (lower). Choice accuracy aggregated by input strength (lines) fits

*Figure 6 continued on next page*

*Figure 6 continued*

well to the empirical data (crosses). The predicted RT aggregated by input strength for correct (solid line) and error (dashed line) trials capture well the RT for correct (filled dots) and error (empty dots) trials in empirical data. (**D**) The model with best-fitting parameters to the behavior replicates the neural dynamic features of the recorded neural activity. *R* unit activities aligned to the onset of stimulus inputs (left) and aligned to the time of model decision (right) replicate the stereotyped ramping dynamics of units associated with the chosen side (solid lines) and suppression of units associated with the unchosen side (dashed lines) under different levels of input strength. The mean activities at the early stage (the smallest median RT of the six conditions, i.e., 410 ms after the onset of the stimulus, indicated by arrows **a** and **b**) and at the onset of model choice (indicated by arrows **c** and **d**) were examined in the following panels. (**E**) Quantification of the best-fit-to-behavior model prediction (dots and lines) to the empirical recordings (crosses). Upper panel: the early-stage activities at the median RT indicated by arrows **a** (chosen side) and **b** (unchosen side). Lower panel: the late-stage activities aligned to the onset of model choice (30 ms before saccade) indicated by arrows **c** (chosen side) and **d** (unchosen side). The model activities were rescaled to the threshold of the empirical activities, i.e., the mean activity across coherences indicated at arrow **c**. The root-mean-square error (RMSE) between the data and the model at the median RT and at the choice onset was calculated and indicated on the panels. (**F**) The model predicted *G* dynamics show a faster decreasing in the chosen units than the unchosen units, indicating that the chosen units are more strongly disinhibited. (**G**) Average *G* unit activity as a function of motion coherence at the time points indicated by letters (see panel **F**) for data aligned to stimulus onset (upper) and to model choice (lower). Activity is sorted by units associated with the chosen (solid lines) and unchosen (dashed lines) sides. (**H**) The model predicted *D* activities ramp in the early (dynamics on the left, sorted to the stimulus onset) and late (dynamics on the right, sorted to the choice onset) stages. (**I**) Average *D* unit activity as a function of motion coherence analogous to panel **G**.

The online version of this article includes the following figure supplement(s) for figure 6:

**Figure supplement 1.** Log-likelihood surfaces for local disinhibition decision model (LDDM) fit to Roitman and Shadlen (2002) behavioral data over the regimes of the seven free parameters.

**Figure supplement 2.** Parameter recovery of local disinhibition decision model (LDDM) for the parameters from the best fit to *Roitman and Shadlen, 2002* behavioral data.

**Figure supplement 3.** Collinearity between self-excitation $\alpha$ and baseline gain control $B_G$.

**Figure supplement 4.** Fit of the original recurrent network model (RNM) to *Roitman and Shadlen, 2002* behavioral data with eight free parameters.

**Figure supplement 5.** Fit of the leaky competing accumulator (LCA) model to *Roitman and Shadlen, 2002* behavioral data, with four free parameters (*Usher and McClelland, 2001*).

best-fitting model, with predicted activity aggregated across trials and aligned to the onset of stimuli and the time of decision as in the original study (*Roitman and Shadlen, 2002*). Aligned to the onset of stimuli (*Figure 6D*, left), neural responses are aggregated by coherence level and eventual choice and truncated at median RT. These data show clear evidence of WTA competition: chosen (solid) and unchosen (dashed) activity traces diverge over time. Moreover, neural activity is stimulus dependent: the dynamics of both chosen and unchosen units ramp at different, coherence-dependent speeds, consistent with empirical findings consistent with an accumulation process. More quantitatively, we examined the relationship between activity and coherence at the specific time point reported in the original work (arrow points **a** and **b**, *Figure 6E*). Model predictions align well with empirical observations: across the three alternative models, the deviation between empirical recordings and model-predicted activity is the smallest for LDDM (quantified by root-mean-square error (RMSE); RMSE$_{LDDM}$ = 2.74 (*Figure 6E*), RMSE$_{RNM}$ = 20.10 (*Figure 6—figure supplement 4E*), and RMSE$_{LCA}$ = 3.92 *Figure 6—figure supplement 5E*).

Aligned to the onset of decision (*Figure 6D*, right), model *R* unit activity near the time of choice shows further evidence of the WTA competition observed in real neurons: the initial divergence between chosen and unchosen activity traces extends into a categorical coding of choice. The relationship between activity and coherence quantitatively replicates the empirical pattern immediately preceding the decision time (*Roitman and Shadlen, 2002*): chosen activity (indicated by arrow **c** in *Figure 6D* and plotted in *Figure 6E*) no longer shows much difference across coherence conditions, while unchosen activity (indicated by **d** in *Figure 6D* and plotted in *Figure 6E*) retains a decrease. Quantification shows that LDDM again best predicted empirical neural activity with data aligned to choice onset (RMSE$_{LDDM}$ = 6.77 [*Figure 6E*; RMSE$_{RNM}$ = 9.35]; [*Figure 6—figure supplement 4E*]; RMSE$_{LCA}$ = 7.51 [*Figure 6—figure supplement 5E*]). Thus, *R* unit activity – in a model with parameters fit only to behavior – replicates the recorded activity of parietal neurons during both initial decision processing and eventual choice selection.

Unlike the RNM and LCA models, the LDDM predicts different dynamics in different subtypes of interneurons (*Figure 6F–I*). The inhibitory (*G*) units selectively code input values and choice but exhibit complex dynamics due to the interplay of feedforward excitation, lateral inputs, and disinhibition.

Early on (dynamics sorted to the left in *Figure 6F* and upper panel in *Figure 6G*), the *G* activities initially increase due to excitatory drive from *R* units. Later on, when the inhibition from *D* units increases (*Figure 6H*), the *G* activities start to decrease. Near the time of choice (dynamics sorted to the right in *Figure 6F* and the lower panel in *Figure 6G*), the chosen *G* units show lower activities than the unchosen side because of stronger inhibition from *D* as an outcome of WTA competition. The dynamics of *D* units rapidly increase in the early stage, driven by excitatory *R* unit activity (dynamics sorted to the left in *Figure 6H*). Dynamics in the late stage (dynamics sorted to the right in *Figure 6H*) show higher activity on the chosen side than the unchosen side as an outcome of WTA competition. Both types of interneurons show different time-dependent patterns of coherence-dependence that likely reflect the complex dynamics of the system and RT-based data aggregation methods (*Figure 6G and H*). While the activities of different interneuron subtypes have not been widely recorded in decision tasks, these new LDDM predictions provide a testbed for future empirical and theoretical investigations.

## The LDDM integrates normalized value coding and WTA choices

While the LDDM separately replicates normalized value coding and WTA dynamics shown in different empirical studies, a key distinguishing feature of the LDDM is that it can capture both phenomena within a single experimental context. Numerous studies using the random-dot motion paradigm show two stages of dynamics: target (action) representation during the pre-motion stage and WTA selection after the go cue following motion stimuli (*Churchland et al., 2008*; *Rorie et al., 2010*). Neural activity in the pre-motion stage shows a characteristic phasic-sustained dynamic response to the presentation of visual cues; rather than purely sensory information, activity during this stage reflects the magnitude and probability of reward associated with the visual cues (*Rorie et al., 2010*). After the go cue, WTA dynamics reflect an integration of motion information and implement a transition from initial value coding to a categorical coding of choice in the late stage of the decision (*Churchland et al., 2008*; *Ding and Gold, 2010*; *Kiani et al., 2008*; *Roitman and Shadlen, 2002*; *Rorie et al., 2010*; *Shadlen and Newsome, 2001*). Studies of economic choice show a similar set of dynamics, a context-dependent valuation, followed by a shift to WTA after a go cue (*Louie et al., 2011*; *Louie et al., 2014*; *Louie and Glimcher, 2010*; *Pastor-Bernier and Cisek, 2011*; *Sugrue et al., 2004*).

Neural dynamics are also observed to be influenced by the number of options, a feature captured by the LDDM. Specifically, the number of options offered to non-human primates has been empirically observed to affect the neural dynamics during both representation and choice (*Basso and Wurtz, 1997*; *Basso and Wurtz, 1998*; *Churchland et al., 2008*). When the choice set is expanded from two options to four options, early representational activity is lower during pre-motion dynamics (*Figure 7A*) and the speed of WTA dynamics slows after motion onset (*Figure 7C*).

Here, we show that the LDDM replicates the impact of the number of options on both early and late empirical neural dynamics during both the representation phases and the WTA phases observed in real neurons. Under four (versus two) options, LDDM *R* unit activity during the representation stage decreases because of increased recurrent inhibition, driven by multiple contextual inputs (left side in *Figure 7D*). Similarly, the ramping speed after motion onset and disinhibition decreases in the four-option (versus the two-option) condition, despite identical parameters (*Figure 7E*). These results highlight the LDDM as a potential mechanism of integrating normalized value coding and WTA competition within a single-circuit architecture.

## Disinhibition controls point versus line attractor dynamics in persistent activity

We next examine the implications of the local disinhibition architecture for another characteristic of decision-related neural firing: persistent activity. In brain areas such as the parietal (*Kiani et al., 2008*; *Kiani et al., 2014*; *Kiani and Shadlen, 2009*; *Roitman and Shadlen, 2002*; *Shadlen and Newsome, 2001*), prefrontal (*Funahashi et al., 1989*; *Fuster and Alexander, 1971*; *Goldman-Rakic, 1995*; *Rigotti et al., 2013*), and premotor cortices (*Pastor-Bernier and Cisek, 2011*), neurons show elevated firing in the absence of stimulus-driven input over intervals of seconds; such persistent activity is thought to underlie working memory and enable decisions based on internally maintained information. In the RNM, recurrent excitation and feedback inhibition preserve categorical choice information after input withdrawal because of the point-attractor dynamics (*Furman and Wang, 2008*; *Wang, 2002*; *Wong*

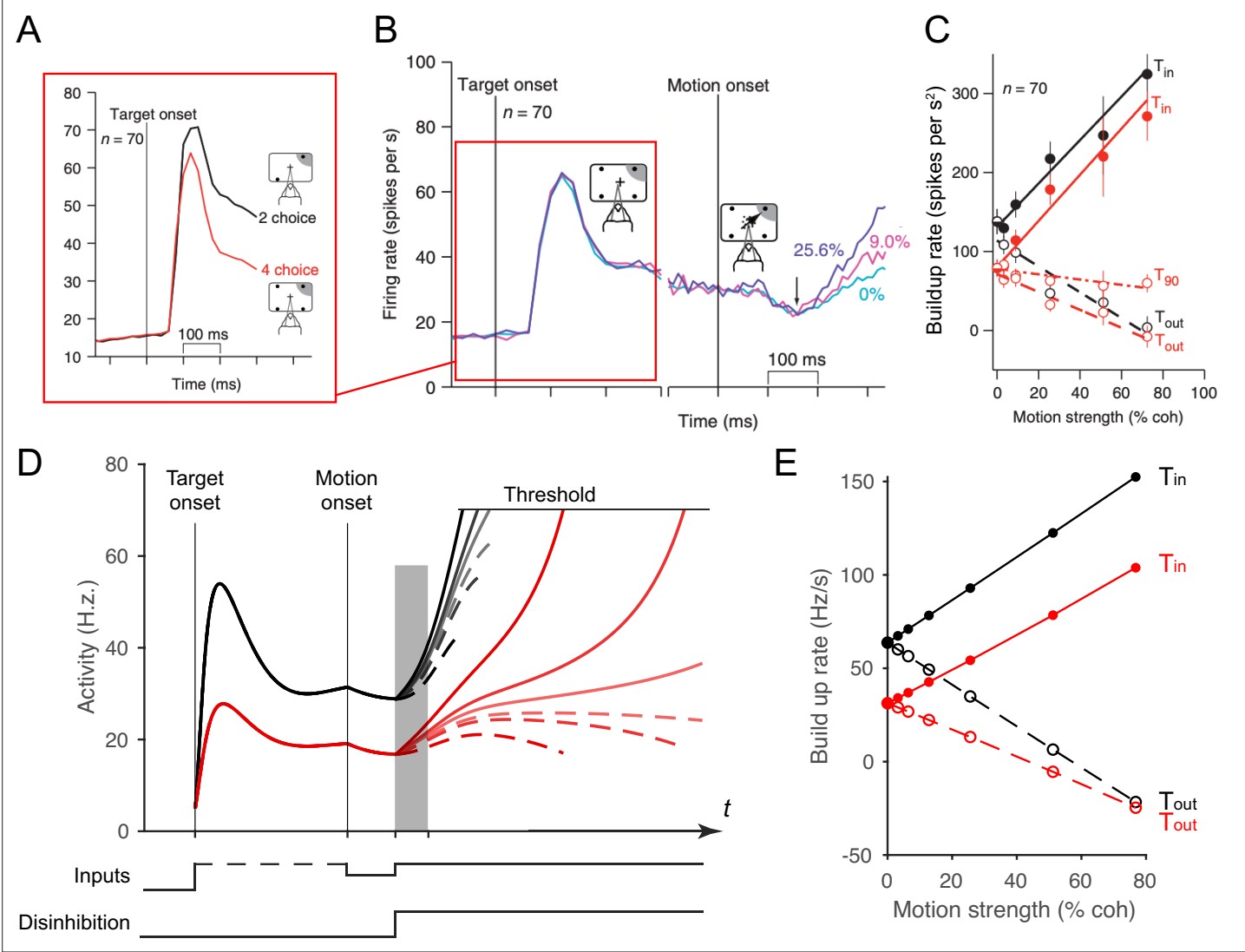

**Figure 7.** Local disinhibition decision model (LDDM) replicates both the normalized coding and winner-take-all (WTA) competition observed sequentially in single neurons examined in a multi-alternative choice task. (**A**) Parietal neuron activity during pre-motion representation is decreased in four-alternative (red) versus two-alternative (black) trials. (**B**) Neural activity during two-alternative choice transitions from pre-motion target representation (left) and to post-motion onset WTA dynamics (right), shown for different input coherences (indicated by colors). (**C**). Ramping speed in two (black) and four (red) alternative conditions, separated for choices toward ($T_{in}$) and away from ($T_{out}$) the neural response field ($T_{90}$ in the original study designates choices for targets orthogonal to the Tin-Tout target, and is not examined here). (**D**) Dynamics of LDDM $R$ unit activity during pre-motion representation without disinhibition (left) and after motion onset with disinhibition (right). (**E**) LDDM replicates the decrease in ramping rates (time period shaded in **D**) from two (black) to four (red) alternatives after the motion onset, consistent with the empirical data.

© 2008, Springer Nature. Panels A-C are reprinted from Figures 3B, 2C and 4F, respectively, from *Churchland et al., 2008*, with permission from Springer Nature. These are not covered by the CC-BY 4.0 license, and further reproduction of these panels would need permission from the copyright holder.

*and Wang, 2006*). Here, we answer two questions: does the LDDM generate persistent activity, and how does this persistent activity differ from that in the RNM?

We found that the LDDM can generate two distinct forms of persistent activity, controlled by the state of disinhibition. *Figure 8A* shows example dynamics of two $R$ units before and after the withdrawal of inputs while disinhibition is silent. Following input withdrawal, network activity decreases but still preserves elevated firing rates, governed by the self-excitation parameter $\alpha$ (the network loses elevated activity when $\alpha \leq 1 + B_G$). The persistent activity ratio between $R_1$ and $R_2$ preserves the ratio between the input values $V_1$ and $V_2$ during the memory interval in contrast to RNMs which

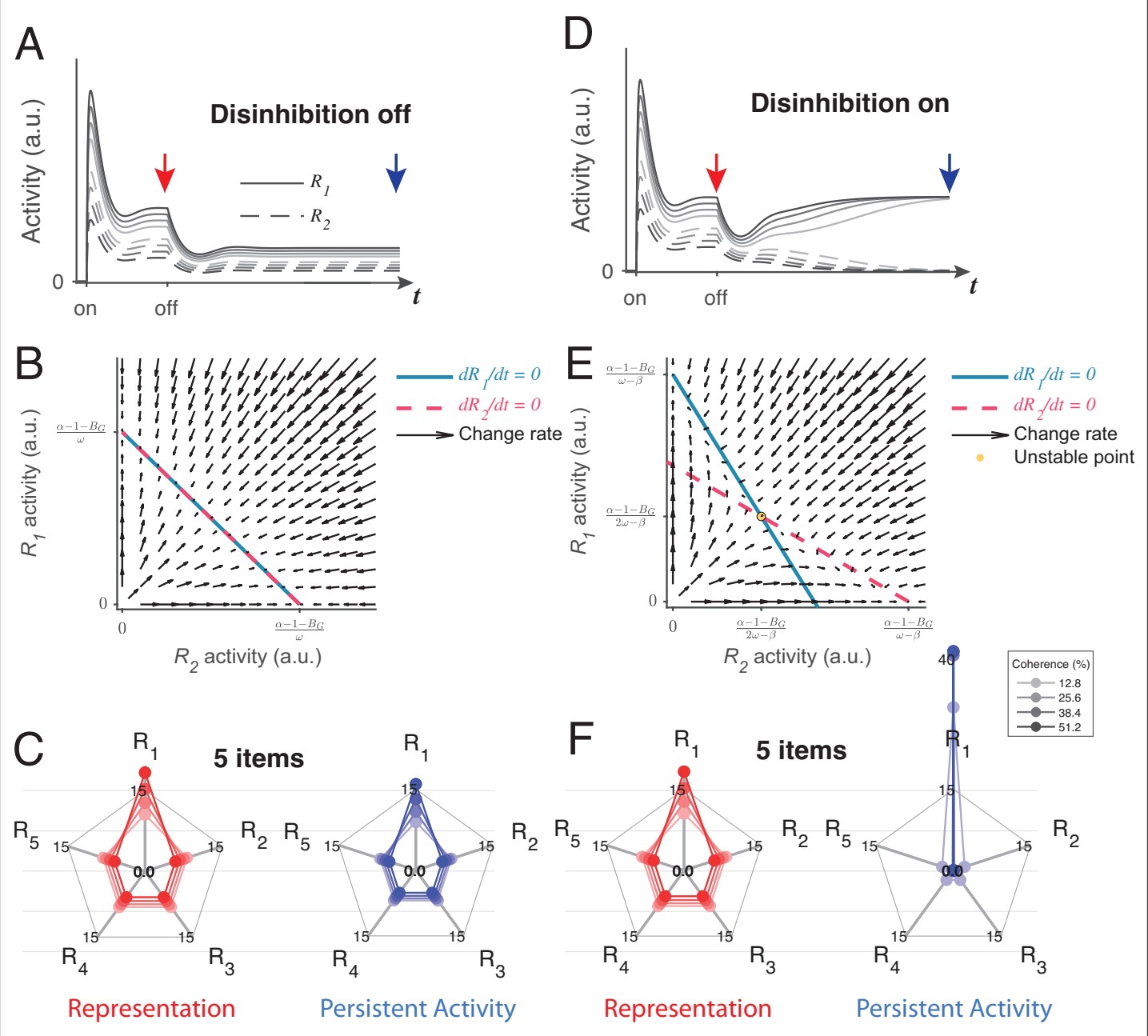

**Figure 8.** Local disinhibition decision model (LDDM) disinhibition controls the flexible implementation of either line attractor or point attractor dynamics in persistent activity. (**A–C**) LDDM under silent disinhibition preserves the input ratio information during persistent activity. (**A**) Example $R_1$ (solid) and $R_2$ (dashed) activities before and after withdrawal of stimuli under different levels of inputs. Neural activity decreases after withdrawal but reaches a new steady that preserves the graded coding of the inputs. (**B**) Phase plane analysis of persistent activity exhibits a line attractor under inactivated disinhibition. The nullclines of $R_1$ (blue) and $R_2$ (red) intersect on the line of attractors, on which the summed value of $R$ activities is a constant ($\frac{\alpha - 1 - B_G}{\omega}$). Red arrows indicate the instantaneous change rate of $R_1$ and $R_2$ at given initial values, following the direction that preserves the $R_1$–$R_2$ ratio. (**D–F**) Persistent activity under active disinhibition preserves only the largest item as categorical information. (**D**) Example $R_1$ and $R_2$ activities before and after withdrawal of stimuli. Disinhibition activates at the same time as the offset of stimuli. During the delay period, the activity dynamic gradually switches from a graded coding of the inputs to a winner-take-all (WTA) type of categorical coding, preserving only the larger item. (**E**) Delay period phase plane analysis exhibits a point-attractor state under activated disinhibition. The nullclines of $R_1$ (blue) and $R_2$ (red) intersect on an unstable point. Red arrows indicating the instantaneous change rate of $R_1$ and $R_2$ bifurcate from the middle to the side corners, resulting in a high-contrast categorical coding. (**C** and **F**) Expansion of the LDDM from a two-item circuit to a five-item circuit, under inactivation and activation of disinhibition. Each axis on the radar plot indicates the activity of one $R$ unit. Dots connected with a line indicate the $R$ activities under the same input conditions. The input values change according to coherence level (**c'**) as $S^*[1+c']$ for $R_1$ and $S^*[1-c']$ for $R_2$ to $R_5$. Representation before the withdrawal of inputs (left panels in

*Figure 8 continued on next page*

*Figure 8 continued*

**C** and **F**) and persistent activity without disinhibition (right panel in **C**) preserve the information about the input values. While persistent activity under disinhibition (right panel in **F**) only preserves the item that received the largest input, with activities of the other items suppressed.

The online version of this article includes the following figure supplement(s) for figure 8:

**Figure supplement 1.** Analysis of local disinhibition decision model (LDDM) persistent activity under generalized gain control weights.

**Figure supplement 2.** Local disinhibition decision model (LDDM) persistent activity under different levels of local disinhibition.

immediately lose all value information and only preserve categorical information about the largest value (see *Figure 8—figure supplement 1* and **Methods** *Analysis for persistent activity* for mathematical proof). Phase plane analyses suggest that relative value coding in persistent activity arises from a line-attractor dynamic in the network during the inactivation of disinhibition, unlike the point-attractor dynamics in the RNM, which shed value information immediately (*Figure 8B*). Like other line-attractor models of persistent activity that store continuous-valued information (*Burak and Fiete, 2009*; *Compte et al., 2000*; *Ganguli et al., 2008*; *Seung, 1996*), an unbiased coding of the input ratio requires perfectly balanced gain control weights from *G* to *R*. Unbalanced weights will result in distorted coding of the input ratio, and graded coding of the inputs will decay over time (*Figure 8—figure supplement 1D and E*). For perfectly balanced weights, the line attractor state is vulnerable to noise perturbation. A small perturbation can easily drive the activity to drift on the line of attractors, with the summed value of $R_1$ and $R_2$ as a constant ($\frac{\alpha-1-B_G}{\omega}$). The preserved ratio between $R_1$ and $R_2$ drifts stochastically over time, similar to the prediction of other line-attractor circuits and consistent with behavioral and neural variability related to working memory (*Seung, 1996*; *Wimmer et al., 2014*).

However, a line attractor is not the only state that the LDDM predicts. If disinhibition is activated during the delay interval, the network switches to a point attractor dynamic similar to the one exhibited by the RNM (see *Figure 8—figure supplement 2* and **Methods** *Analysis for persistent activity* for mathematical proof). *Figure 8D* shows the example dynamics of two *R* units before and after the withdrawal of inputs. Disinhibition drives a competition between the two *R* units, resulting in a switch between the graded coding of the input ratio to a categorical coding of the largest value ($\beta = .4$ in visualization). Interestingly, a transition of coded information from input values to categorical information has been widely observed in firing rates in decision-related regions, such as LIP and the superior colliculus, during the delay period of decision-making (*Rorie et al., 2010*; *Shadlen and Newsome, 2001*; *Zhang et al., 2021*). The point attractor predicted by the circuit under disinhibition (*Figure 8E*) is highly tolerant to perturbations compared to the line attractor. Choice performance over long delays may require a switch from the value coding to the categorical regimes to achieve this robustness. As a plausible biological mechanism for mediating top-down control, disinhibition may gate such a transition without imposing any distinct change on the network architecture.

The LDDM can be easily expanded to multiple options. Here, we show an example of a five-option case with five sets of option-specific *R-G-D* units. A line attractor network with silent disinhibition (*Figure 8C*, right) is able to retain relative input value information for all five items simultaneously in the network. Due to normalization, the neural activity representing each alternative decreases with the total number of alternatives, with the summed value as a constant ($\sum_1^N R_i = \frac{\alpha-1-B_G}{\omega}$), leading to a lower signal-to-noise ratio when coding more items; this set-size effect may be related to working memory (WM) span constraints (*Cowan, 2010*; *Cowan, 2016*; *Engle, 2001*; *Engle, 2002*; *Oberauer et al., 2016*). When disinhibition is active, the LDDM exhibits a point attractor (*Figure 8F*, right), and the network only holds the information of the largest item as a categorical code during persistent activity.

## Gated disinhibition provides top-down control of choice dynamics

In addition to its crucial role in generating WTA competition, local disinhibition provides an intrinsic mechanism for top-down control of choice dynamics. Decision circuits show remarkable flexibility in timing, with similar neurophysiological evidence of this flexibility recorded in a variety of task paradigms. In addition to reaction-time tasks, in which subjects can choose at any time immediately after the onset of stimulus, decision-related neural activity has been widely studied in fixed-duration and delayed-response tasks. In fixed-duration tasks, subjects are required to withhold their selection of action until an instruction signal. Neural activity prior to the instruction signal reflects value information,

for example, about reward characteristics (*Dorris and Glimcher, 2004*; *Louie et al., 2011*; *Platt and Glimcher, 1999*; *Sugrue et al., 2004*; *Watanabe, 1996*) or accumulating perceptual evidence (*Kiani et al., 2008*; *Kiani et al., 2014*; *Kiani and Shadlen, 2009*; *Kim and Shadlen, 1999*; *Roitman and Shadlen, 2002*; *Rorie et al., 2010*; *Shadlen and Newsome, 2001*); however, this activity never entirely diverges or reaches the decision threshold until after the instruction cue, suggesting a gating of the competition process. In delayed-response (working memory) tasks, subjects must postpone selection for an interval that includes both stimulus presentation and an additional subsequent interval after the stimulus is withdrawn. As in fixed-duration tasks, neural activity in delayed-response tasks typically carries decision–related information (across both the stimulus and delay periods), but WTA selection – and behavioral choice – is withheld until the instruction cue is given (*Kiani et al., 2008*; *Kiani et al., 2014*; *Kiani and Shadlen, 2009*; *Kim and Shadlen, 1999*; *Roitman and Shadlen, 2002*; *Shadlen and Newsome, 2001*). Thus, biological decision circuits are able to evaluate choice options while selectively initiating the WTA selection process with variable context-dependent timing.

Despite this evidence of top-down control, how neural circuits implement dynamic control of selection – and temporal separation of evaluation and WTA choice – is largely unaddressed in current decision models. For example, in RNM models, neural activity is driven by fixed attractor dynamics; option evaluation and the selection process cannot be disambiguated, and WTA competition is essentially ballistic and not under top-down control. In this section, we examine how the timing of a dynamic top-down control signal – modulating the strength of disinhibition via long-range inputs and neuro-modulation – allows the LDDM to capture neural activity in different task paradigms. In these simulations, disinhibition is activated when the choice instruction cue is presented. *Figure 9A* shows LDDM activity in a reaction-time task, a standard paradigm in the perceptual decision-making (*Churchland et al., 2008*; *Roitman and Shadlen, 2002*). As in previous analyses (*Figures 5 and 6*), LDDM *R* units show simultaneous evaluation (coherence-dependent ramping) and WTA selection (rise to threshold) processes driven by immediate activation of disinhibition at motion stimulus onset.

In a fixed-duration task (*Figure 9B*), disinhibition is activated after a required interval of stimulus presentation as in the empirical data. Compared to the reaction-time task, LDDM activity here shows distinct, temporally separated patterns during stimuli viewing and option selection; this temporal segregation is driven by the activation of disinhibition (a step function on $\beta$ in this example), which promotes a transition between value representation and WTA choice.

A further demonstration of this temporal flexibility arises from considering delayed-response tasks (*Figure 9C*), which include an interval between stimuli offset and the onset of the instruction cue. Consistent with its ability to maintain persistent activity (*Figure 8*), the LDDM shows value coding across the delay interval. It delays WTA selection until after the instruction cue and the accompanying activation of disinhibition. These results show that the LDDM – via modulation in the timing of disinhibition activation - can temporally separate the value representation and selection processes (unlike the RNM), enabling it to capture the diversity of neural dynamics seen in reaction-time, fixed-duration, and delayed-response tasks.

## Inhibitory potentiation distinguishes LDDM from earlier models

The architecture of disinhibition employed by the LDDM is more structured than the earlier non-selective inhibition used in most standard competition networks. This distinction gives rise to the novel prediction from the LDDM that the influence of global changes in inhibitory tone is non-selective during representation but switches to input-selective after disinhibition is increased. This reflects a fundamentally novel prediction of this class of model. The LDDM contains two different types of inhibition, and thus, its reaction to inhibitory potentiation depends on both the state of the disinhibitory network and the intensity of potentiation. To highlight the importance of that prediction, we implemented different levels of inhibitory connection weights in both the LDDM and the standard RNM.

At the neural level, the LDDM predicts a dissociable effect of potentiated inhibition on the primary (*R*) neuron's activity (*Figure 10A*). During option representation (cue interval in fixed duration trials), potentiated inhibition increases both recurrent and lateral inhibition, leading to decreased firing rates and a weaker modulation by value in the *R* neurons. During option selection (go/choice intervals in fixed duration trials), local disinhibition increases WTA activity and decreases the late-stage representation of value. As an outcome, these changes produce a speeding up of RTs but a reduced choice accuracy (*Figure 10B*). The expected differences between the control condition and the inhibitory

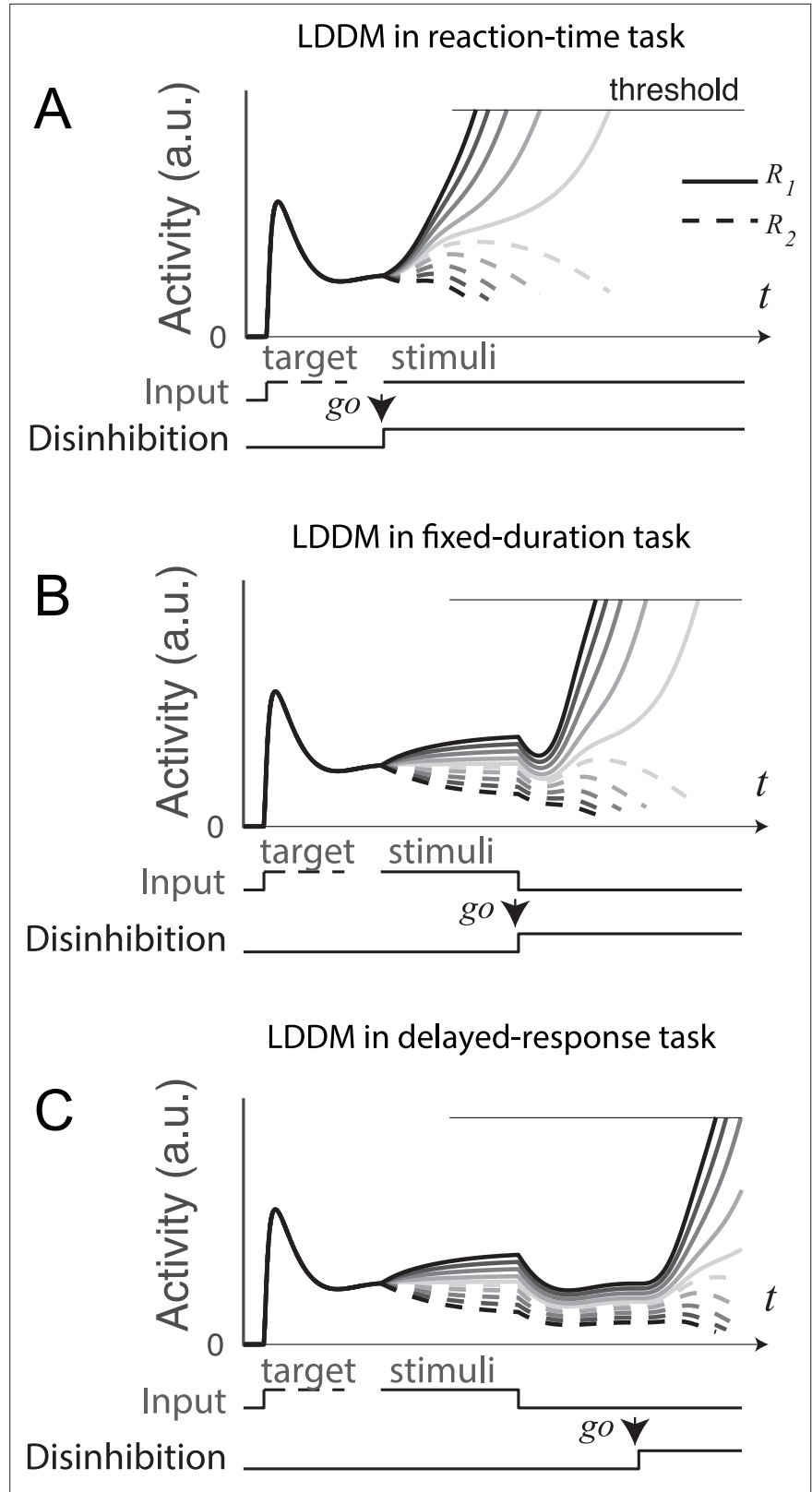

**Figure 9.** Gated disinhibition flexibly adapts the dynamics of the circuit to various types of tasks. All of the tasks consist of a pre-stimulus stage with equal inputs to $R_1$ (solid) and $R_2$ (dashed) and a stimulus stage with input values determined by the stimuli (indicated by grayscale, the same value matrix as used in **Figure 5A**). (**A**) Reaction-time task. Subjects are free to respond at any time following stimulus onset, and model disinhibition is activated

*Figure 9 continued on next page*

*Figure 9 continued*

with the onset of stimuli. Model dynamics show winner-take-all (WTA) competition right after the onset of stimuli. (**B**) Fixed duration task. Subjects are required to wait for a fixed duration of stimulus viewing before choice, and model disinhibition is turned on only at the onset of the instruction cue (usually indicated in experiments by fixation point offset). Model dynamics show normalized value coding before the instruction cue and a transition to WTA choice afterward. (**C**) Working memory (delayed response) task. Subject choice occurs after an interval of stimulus presentation and a subsequent delay interval without stimuli, and model disinhibition is turned on at the end of the delay period. Model dynamics exhibit normalized value coding during stimulus input, preserved relative value information during the delay period, and a transition to WTA choice dynamic after the instruction cue.

potentiation condition would be evident in chronometric and psychometric curves across different levels of inputs effectively implementing a speed-accuracy tradeoff (*Figure 10C*). Note that the qualitative predictions for inhibitory potentiation effects on RT and accuracy are robust to specific LDDM parameterizations (*Figure 10D*). In contrast, in more traditional networks like the RNM that employ non-selective inhibition, potentiated inhibition suppresses the excitatory neural activities during the WTA competition (*Figure 10E*). The suppression in neural coding in these models slows down RTs but does not affect choice accuracy (*Figure 10F and G*), thus failing to replicate the observed speed-accuracy tradeoff. We note that these novel predictions that differentiate models which rely on structured disinhibition could be readily tested using modern optogenetic techniques.

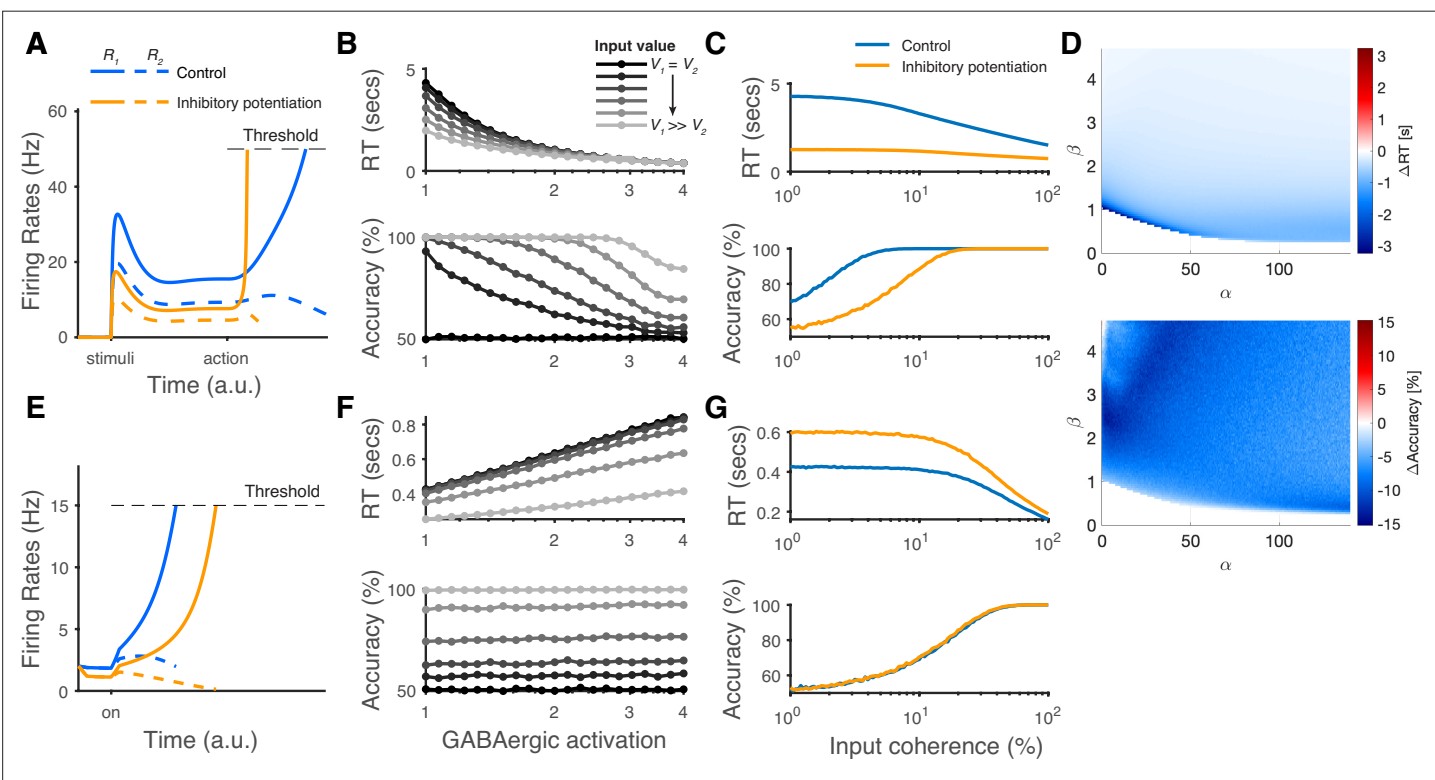

**Figure 10.** The modeling predictions of inhibitory potentiation to decision-making neural dynamics and behaviors. (**A**) The predicted neural dynamics of pyramidal neurons ($R_1$, solid lines, and $R_2$, dashed lines) activities in a fixed duration decision task from local disinhibition decision model (LDDM). The inhibitory potentiation condition (orange) compared to the control condition (blue) decreases neural activities during early-stage representation but speeds up winner-take-all (WTA) bifurcation during choice. (**B**) Increasing the levels of Inhibitory potentiation speeds up RTs but decreases choice accuracy, examined over multiple levels of input coherences (indicated by grayscales). (**C**) Comparing inhibitory potentiation (orange) with control (blue), the differences will be evident in average chronometric and psychometric curves. (**D**) The predicted behavioral pattern can be generalized across the full space of $\alpha$ and $\beta$ parameters regime in the LDDM. (**E**) The predicted neural dynamics of primary neurons ($R_1$, solid lines, and $R_2$, dashed lines) activities from recurrent network models (RNMs; e.g. *Wong and Wang, 2006*). Since the model does not include a mechanism of switch, the fixed duration task is not able to be tested in this type of model. We examined the reaction time task instead. RNM predicts suppressed neural dynamics under inhibitory potentiation. (**F**) RNM predicts increased RTs but unchanged accuracy. (**G**) The chronometric and psychometric curves predicted by RNM will be qualitatively different from LDDM.

## Discussion

The prevalence of disinhibitory circuit motifs in the brain, and recent evidence for structured decision-related inhibitory activity, argue for a more structured implementation of inhibition than has been previously employed in computational models of decision-making. Here, we show that the disinhibition-based LDDM replicates three characteristic features of observed neurobiological decision-making circuits – normalized value coding, WTA choice, and persistent activity – within a single-circuit architecture. We find that our disinhibition-based model outperforms existing recurrent circuits both in fitting empirical choice data and in replicating decision-related neural dynamics. Perhaps most importantly, the LDDM provides a novel mechanism for top-down control of decision dynamics which regulates phenomena like the empirically observed speed-accuracy tradeoff. By controlling the timing of disinhibition, the LDDM effectively paces the decision process and replicates neural dynamics from a broader range of empirical choice tasks than any previous models.

### Flexible control of dynamic regimes

While normalized value coding and WTA selection have largely been modeled separately, the LDDM offers a biologically plausible circuit architecture that integrates these two features via local disinhibition. Existing neurophysiological evidence shows that WTA dynamics and normalized value coding co-exist in the same brain regions. On the one hand, neural activities show relative value coding in the early stage of decision-making, reflecting a context-dependent modulation consistent with the canonical divisive normalization computation (*Churchland et al., 2008*; *Kira et al., 2015*; *Louie et al., 2011*; *Pastor-Bernier and Cisek, 2011*; *Rorie et al., 2010*; *Strait et al., 2014*; *Yamada et al., 2018*). On the other hand, WTA choice dynamics are widely observed during later stages of decision-making across multiple brain regions of non-human primates (*Andersen and Buneo, 2002*; *Churchland et al., 2008*; *Ding and Gold, 2010*; *Ding and Gold, 2012*; *Ding and Gold, 2013*; *Dorris and Glimcher, 2004*; *Hanks et al., 2014*; *Kiani et al., 2008*; *Kiani et al., 2014*; *Kim and Shadlen, 1999*; *Louie and Glimcher, 2010*; *Padoa-Schioppa, 2013*; *Padoa-Schioppa and Conen, 2017*; *Pastor-Bernier and Cisek, 2011*; *Platt and Glimcher, 1999*; *Roesch and Olson, 2003*; *Roitman and Shadlen, 2002*; *Rorie et al., 2010*; *Shadlen and Newsome, 2001*; *Sugrue et al., 2004*; *Thura and Cisek, 2014*; *Yamada et al., 2018*), including many of the brain regions that show normalized value coding. In addition, neural firing rates show a graded coding of perceptual evidence and reward during the early stage of decision-making tasks that require evidence accumulation, gradually transitioning to a categorical coding for choice in the late period of decision-making (*Churchland et al., 2008*; *Dorris and Glimcher, 2004*; *Gold and Shadlen, 2007*; *Platt and Glimcher, 1999*; *Roitman and Shadlen, 2002*; *Rorie et al., 2010*; *Shadlen and Newsome, 1996*; *Sugrue et al., 2004*; *Zhang et al., 2021*).

All existing models of decision-making capture activity dynamics only in specific temporal intervals during decision-making tasks or across trials in specific task paradigms (*Hart and Huk, 2020*; *Hunt et al., 2012*; *Louie et al., 2014*; *Wang, 2002*; *Wong and Wang, 2006*), and thus, typically do not generalize across tasks in the same way as the empirically observed neural architecture. In contrast, the LDDM presented here modulates the dynamics of the circuit without requiring changes in circuit structure via gated disinhibition driven by the external action instruction. Controlling the timing of valuation-to-WTA regime transition enables the LDDM to replicate neural dynamics in a much more diverse set of task paradigms with different stimulus and action timing schedules (*Kiani et al., 2008*; *Roitman and Shadlen, 2002*; *Rorie et al., 2010*; *Shadlen and Newsome, 2001*).

### Biological plausibility and fast modulation of disinhibition

The top-down control of normalization via disinhibition used in the model mirrors recently proposed mechanisms for flexible modulation of contextual processing in sensory circuits (*Coen-Cagli et al., 2012*; *Coen-Cagli et al., 2015*; *Schwartz and Coen-Cagli, 2013*). The input-scaled disinhibition we employ implements a self-sparing ('donut-like') inhibition motif central to existing midbrain models of categorical selection (*Mahajan and Mysore, 2022*; *Mysore and Kothari, 2020*). The micro-circuit structure underlying this donut-like inhibition has been revealed as a mechanism of localized disinhibition from VIP neurons to PV/SST neurons in the cortex (*Karnani et al., 2016*). Recent research on neuromodulatory control of disinhibition offers biologically plausible mechanisms for such top-down control of circuit dynamics. In addition to evidence that VIP neurons are recruited by long-range projections from distant regions (*Lee et al., 2013*; *Zhang et al., 2014*), VIP neurons are recruited

by neuromodulatory projections such as acetylcholine (*Fu et al., 2014*) from the basal forebrain and pedunculopontine nuclei and serotonin from the red nucleus. With ionotropic acetylcholine receptor (nAChR) and serotonin receptors ($5HT_{3a}R$ and $5HT_2R$), VIP neurons depolarize to acetylcholine and serotonin (*Alitto and Dan, 2012*; *Pfeffer et al., 2013*; *Rudy et al., 2011*; *Tremblay et al., 2016*). The spiking mode of a major type of VIP neurons in layer II/III of the cortex switches from an input-insensitive burst-quiescent mode to an input-sensitive tonic mode under the cholinergic and serotonin modulation (*Prönneke et al., 2020*). Such a mode-switching feature allows the disinhibitory neurons to receive excitatory projections with different gains under different levels of neuromodulation, providing a mechanism to modulate network dynamics via disinhibition without a change in network structure that we employ as a central feature of the LDDM. In vivo studies show that disinhibition mediated by cholinergic activation is triggered in a surprisingly fast time scale of tens of milliseconds (*Alitto and Dan, 2012*; *Hangya et al., 2015*; *Letzkus et al., 2011*), supporting a fast modulation mechanism of disinhibition and network plasticity of the kind the LDDM instantiates.

## The contribution of LDDM relative to existing disinhibition models

Disinhibition has been previously linked in separate models to several of the computational functions that are exhibited in a unified manner by the LDDM. For example, a computational model employing dendritic disinhibition captures flexible information routing in a context-dependent decision task, with dendritic disinhibition gating on specific inputs to a circuit while gating off other pathways (*Yang et al., 2016*). However, disinhibition plays a different role in this model (context-dependent input gating) from that employed in the LDDM (transition from value coding to WTA selection and mutual competition). In another example, PV neuron activation within a disinhibitory circuit motif can produce a divisive normalization of tuning curves in a model of the visual cortex (*Litwin-Kumar et al., 2016*). This specific model of division, however, arises from different circuit mechanisms than those we employ, such as reduced tuned input and firing rate nonlinearities. Finally, disinhibition has also been proposed to underlie the long time scales of information processing seen in working memory, as enhancing inhibitory-to-inhibitory connections stabilize temporal dynamics and improve working memory performance in recurrent neural networks (*Kim and Sejnowski, 2021*). One other notable difference between previous research and our current work is that disinhibition in past models typically contributes to a specific function (e.g. input gating, categorical selection, working memory, etc.), whereas disinhibition in the LDDM both mediates a transition from value coding to WTA selection and plays an integral role in the selection process itself. Taken together, previous results and our current work reinforce the importance of incorporating disinhibition in circuit models of decision-making.

## Disinhibition in cortical-ganglia pathways: similarities and the differences

While largely absent in standard existing cortical decision models, disinhibition is a key element of action selection in models of the cortical-basal ganglia (CBG) system (*Bogacz and Gurney, 2007*; *Frank, 2005*; *Lo and Wang, 2006*; *Schroll and Hamker, 2013*; *Wei et al., 2015*). In the basal ganglia direct pathway, GABAergic neurons in the striatum inhibit neurons in the substantia nigra pars reticulata and internal globus pallidus, which in turn send inhibitory projections to the thalamus. Cortical inputs to the striatum thus produce a disinhibition of thalamic outputs to the cortex and brainstem motor areas, resulting in motor facilitation. Crucially, the activation of disinhibition in the CBG system is selective: the selection of a specific action requires a selective disinhibition driven by asymmetries in cortical inputs or striatal synaptic weights. This selective disinhibition is an essential element of computational models of the CBG system (*Frank, 2005*; *Lo and Wang, 2006*), including more complex models that incorporate global inhibition mediated by the indirect and hyper-direct pathways (*Bogacz and Gurney, 2007*; *Schroll and Hamker, 2013*; *Wei et al., 2015*).

While both the LDDM and standard CBG models utilize disinhibition to drive selection, they differ in two important ways. First, disinhibition in the LDDM specifically functions to implement a transition between value coding and WTA selection states. This transition is mediated by a broad/non-selective activation of disinhibition across the decision circuit. The activation of disinhibition is not biased toward specific alternatives until a period of interaction with differential value inputs to option-specific subcircuits that instantiates the WTA process. Second, disinhibition in the LDDM is tightly integrated with the lateral inhibition that mediates competition (and hence normalization) between alternatives;

consistent with the microarchitecture of the cortex which it seeks to model (*Fu et al., 2014*; *Karnani et al., 2016*; *Kepecs and Fishell, 2014*; *Pi et al., 2013*; *Zhang et al., 2014*), disinhibitory, inhibitory, and excitatory neurons are part of the same local circuit. In contrast, the basal ganglia are known to lack these local, lateral connections and mutual competition. As a result CBG models typically require both direct pathway disinhibition along with diffusive suppression of competing motor plans via the indirect or hyper-direct pathways (*Bogacz and Gurney, 2007*; *Schroll and Hamker, 2013*; *Wei et al., 2015*) for effective operation. Thus, while conceptually similar to the CBG models, disinhibition in the LDDM is in some ways quite distinct, being tightly integrated with competitive inhibition and providing dynamic control of circuit state, both characteristics of decision-making in cortical brain areas.

## Point- and line-attractor persistent activity

An interesting feature of the LDDM is that it can produce both point attractor (*Bathellier et al., 2012*; *Kopec et al., 2015*; *Niessing and Friedrich, 2010*; *Wills et al., 2005*) and continuous/line attractor (*Ganguli et al., 2008*; *Wimmer et al., 2014*; *Yoon et al., 2013*) dynamics in persistent activity, the balance between these two being controlled by the level of disinhibition. Given ambiguous empirical evidence, it remains controversial whether persistent activity in neural circuits exhibits point attractor (*Bathellier et al., 2012*; *Kopec et al., 2015*; *Niessing and Friedrich, 2010*; *Wills et al., 2005*) or continuous/line attractor (*Ganguli et al., 2008*; *Wimmer et al., 2014*; *Yoon et al., 2013*) dynamics. Most existing circuit models of persistent activity exclusively predict either a point attractor (*Amit and Brunel, 1997*; *Brunel and Wang, 2001*; *Hopfield, 1982*; *Wang, 1999*) or a line attractor (*Amari, 1977*; *Burak and Fiete, 2009*; *Compte et al., 2000*; *Ganguli et al., 2008*; *Seung, 1996*). The LDDM achieves the flexible reconfiguration of line attractor and point attractor states under the control of disinhibition, suggesting that attractor dynamics might not be a fixed property of a network; rather, it may be adaptive and controllable by a top-down signal operating via gated disinhibition. Of course, similar reconfiguration has been achieved by other important circuit mechanisms that have been well-described. For example, a mutual inhibition network can capture the different regimes of sequential two-interval decision-making – stimulus loading, working memory, and comparison – by assuming a flexible reconfiguration of the external inputs (*Machens et al., 2005*). Similar to the LDDM, this model can transition between point attractor (initial stimulus encoding), line attractor (working memory), and saddle point (comparison) dynamics. Interestingly, disinhibition may also play a role in this model by providing a theoretical mechanism to switch the routing of external inputs within the circuit, which drives the switch from line attractor to comparison dynamics.

A second point relevant to persistent activity is that the exact degree of recurrent excitation in the network (controlled by $\alpha$) is unable to be identified from the current datasets owing to its collinearity with the degree of baseline gain control from SST/PV neurons to the pyramidal neurons (controlled by $B_G$). We believe that this feature reflects the E-I balance in the network: with larger recurrence than gain control, the network is able to generate persistent activity when excitatory input is withdrawn; otherwise, the network is unable to maintain such excitability. Since $\alpha$ and $B_G$ are highly collinear in predicting either neural dynamics or behavior, future empirical work is needed to identify the features that dissociate the two parameters. For example, one possible approach is to measure the neural activity of different neuronal types, taking advantage of the advanced genetic labeling and in vivo calcium imaging (*Najafi et al., 2020*). Since we propose that baseline gain control is linked to the activity of SST/PV interneurons, a direct test can be measuring the activities of SST and PV neurons across the full dynamics of decision-making tasks; the identification of $B_G$ will help dissociate its contribution from that of recurrence.

## Conclusions

In conclusion, we introduce a novel, biologically plausible architecture for decision-making based on local disinhibition. Our model unifies the characteristic decision-making features of normalized value coding, WTA competition, and persistent activity in a single circuit. The LDDM captures both psychometric and chronometric aspects of behavioral choice, as well as realistic neural dynamics in essentially all standard decision-making tasks. The local disinhibition it employs provides a mechanism for top-down control of local decision circuit dynamics, enabling the LDDM both to replicate variable task-dependent timing in diverse decision-making paradigms and to implement realistic speed-accuracy

tradeoffs. These results suggest a new circuit mechanism for decision-making that can capture a large suite of empirical data and emphasize the importance of interneuron diversity, local circuit architecture, and top-down control in models of the decision process.

## Methods
### Equilibria and stability analysis of the LDDM

In *Figures 3 and 5*, we showed that the LDDM exhibits different patterns of equilibria and stabilities under normalized value coding and WTA competition, mediated through disinhibition. Here, we provide detailed mathematical analysis about the equilibria and stability of this dynamic system under different states of disinhibition.

Equilibria of the system were solved by taking the intersection of the nullclines of all units, i.e., the steady states of each unit. This is obtained by setting $dR_i/dt$, $dG_i/dt$, and $dD_i/dt$ all equal to 0 in *Equations 1–3*. The solution of the equilibrium state of $R$ units ($R_i^*$) can be written as:

$$R_i^* = \frac{V_i + B_R}{1 + B_G - \alpha + (\omega - \beta) R_i^* + \omega \sum_{j \neq i} R_j^*}$$

(5)

For a binary input system ($N=2$), the six differential equations can be simplified to two equations with only the $R$ units explicitly in the expression (*Equation 6*). Each equation describes the nullcline of a single $R$ unit.

$$\begin{cases} \frac{V_1 + B_R}{R_1^*} - (\omega - \beta) R_1^* - (1 + B_G - \alpha) = \omega R_2^* \\ \frac{V_2 + B_R}{R_2^*} - (\omega - \beta) R_2^* - (1 + B_G - \alpha) = \omega R_1^* \end{cases}$$

(6)

Given that the equilibrium states of the system can be reduced with only $R$ units explicitly in the expression, these equilibrium points can be visualized in the $R_+^2$ space of $R_1$ and $R_2$ activities as the intersection of the nullclines of the two $R$ units (as shown in *Figures 3 and 5*). The stability of each equilibrium point was then examined by checking the eigenvalues of the Jacobian matrix around it. The equilibrium point is attractive and stable when all of the eigenvalues have negative real parts; the equilibrium point is divergent and unstable when there exist any positive real parts of eigenvalues. By denoting $\mathbf{F} = \left(F_{R_1}, F_{G_1}, F_{D_1}, F_{R_2}, F_{G_2}, F_{D_2}\right)$ as the differential equations for all units in their steady states, the Jacobian matrix around the point can be written as *Equation 7*:

$$J = \begin{bmatrix} \frac{\partial F_{R_1}}{\partial R_1} & \frac{\partial F_{R_1}}{\partial G_1} & \frac{\partial F_{R_1}}{\partial D_1} & \frac{\partial F_{R_1}}{\partial R_2} & \frac{\partial F_{R_1}}{\partial G_2} & \frac{\partial F_{R_1}}{\partial D_2} \\ \frac{\partial F_{G_1}}{\partial R_1} & \frac{\partial F_{G_1}}{\partial G_1} & \frac{\partial F_{G_1}}{\partial D_1} & \frac{\partial F_{G_1}}{\partial R_2} & \frac{\partial F_{G_1}}{\partial G_2} & \frac{\partial F_{G_1}}{\partial D_2} \\ \frac{\partial F_{D_1}}{\partial R_1} & \frac{\partial F_{D_1}}{\partial G_1} & \frac{\partial F_{D_1}}{\partial D_1} & \frac{\partial F_{D_1}}{\partial R_2} & \frac{\partial F_{D_1}}{\partial G_2} & \frac{\partial F_{D_1}}{\partial D_2} \\ \frac{\partial F_{R_2}}{\partial R_1} & \frac{\partial F_{R_2}}{\partial G_1} & \frac{\partial F_{R_2}}{\partial D_1} & \frac{\partial F_{R_2}}{\partial R_2} & \frac{\partial F_{R_2}}{\partial G_2} & \frac{\partial F_{R_2}}{\partial D_2} \\ \frac{\partial F_{G_2}}{\partial R_1} & \frac{\partial F_{G_2}}{\partial G_1} & \frac{\partial F_{G_2}}{\partial D_1} & \frac{\partial F_{G_2}}{\partial R_2} & \frac{\partial F_{G_2}}{\partial G_2} & \frac{\partial F_{G_2}}{\partial D_2} \\ \frac{\partial F_{D_2}}{\partial R_1} & \frac{\partial F_{D_2}}{\partial G_1} & \frac{\partial F_{D_2}}{\partial D_1} & \frac{\partial F_{D_2}}{\partial R_2} & \frac{\partial F_{D_2}}{\partial G_2} & \frac{\partial F_{D_2}}{\partial D_2} \end{bmatrix}$$

$$= \begin{bmatrix} -1 + \frac{\alpha}{1+G_1^*} & -\frac{V_1 + B_R + \alpha R_1^*}{\left(1 + G_1^*\right)^2} & 0 & 0 & 0 & 0 \\ \omega & -1 & -1 & \omega & 0 & 0 \\ \beta & 0 & -1 & 0 & 0 & 0 \\ 0 & 0 & 0 & -1 + \frac{\alpha}{1+G_2^*} & -\frac{V_2 + B_R + \alpha R_2^*}{\left(1 + G_2^*\right)^2} & 0 \\ \omega & 0 & 0 & \omega & -1 & -1 \\ 0 & 0 & 0 & \beta & 0 & -1 \end{bmatrix}$$

(7)

We examined the configuration of nullclines and checked the eigenvalues of the Jacobian matrix across a wide range of parameter values $\alpha$ and $\beta$. $\omega$ was set as a unit value of 1 for the sake of simplicity. $B_R$ and $B_G$ were set as zero in the following visualization.

The property of the system under equivalent inputs is a critical test since it determines whether the system is able to implement a WTA choice and select an option. Thus, we examined the property of the system for WTA under equal inputs. Examining the full space of $\alpha$ and $\beta$ revealed five territories

distinguished by the number of equilibrium points and their stabilities (*Figure 5—figure supplement 1A*). For each territory, the configuration of nullclines is illustrated in *Figure 5—figure supplement 1* labeled by color. **Dark green region**: when disinhibition is smaller ($\beta < 1$), $\alpha$ and $\beta$ show a trade-off in generating WTA competition. When both $\alpha$ and $\beta$ are small, the system generates a unique equilibrium point of normalized coding (dark green region in *Figure 5—figure supplement 1A*, nullclines shown in *Figure 5—figure supplement 1B*). Eigenvalues in this regime show all negative real parts on this equilibrium point, indicating it is a stable equilibrium. **Blue region**: as $\alpha$ values increase (at smaller $\beta$ values), the system generates three equilibrium points (*Figure 5—figure supplement 1D*), with two high-contrast (stable) attractors at the peripheral and one (unstable) repellor in the center of space $R_1$–$R_2$. Neural activities of $R_1$ and $R_2$ with equal initial values bifurcate into the high-contrast attractors to realize WTA competition (example traces shown in red and blue lines). **Green region**: when the strength of disinhibition increases ($\beta > 1$), most of the regimes (yellow and red regions) show the properties of WTA competition except for a small regime when $\alpha < 1 + B_G$ (an almost invisible region between dark green and yellow). In the green region, the nullclines of $R_1$ and $R_2$ still intersect on three equilibrium points, but in contrast to the blue region, the two points with a high contrast of $R_1$–$R_2$ activities are unstable and the equilibrium point in the center is stable; therefore, the system maintains normalized coding (*Figure 5—figure supplement 1C*). **Yellow region**: when disinhibition is large ($\beta > 1$), most of the parameter regime in the yellow region shows only one repellor at the center (*Figure 5—figure supplement 1E*). The activities of $R_1$ and $R_2$ bifurcate from the center repellor to the high-contrast corners. The restriction of maximum activity depends on the value of $\alpha$. When $< 1 + B_G$, the model predicts a limited value of activity on each $R$ unit as ($\frac{V_i + B_R}{1 + B_G - \alpha}$; vertical and horizontal dashed lines in *Figure 5—figure supplement 1E*). When $\alpha \geq 1 + B_G$, the model predicts no boundary on the maximum activities (though a boundary may still need to be considered because of biological constraints). **Red region**: when disinhibition is extremely large ($\beta > 2$), the two nullclines show no intersections (*Figure 5—figure supplement 1F*). Most of the other features in this region are similar to the yellow region. The neural activities of $R_1$ and $R_2$ bifurcate from initial values from the center to the corners of high contrast (example traces shown in red and green thin lines). The boundary of neural activity is predicted when $\alpha < 1 + B_G$ and not accounted when $\alpha \geq 1 + B_G$.

Taken together, the five territories can be simplified into two regions based on the properties of the system in implementing either normalized coding or WTA competition as discussed in the main text (*Figure 5E*). These two regions show clear-cut dichotomous separation in the two-dimensional space of recurrent excitation weight ($\alpha$) and local disinhibition weight ($\beta$).

## Numerical simulations

To quantify neural dynamics and behavioral performance (choice/RT), time-varying activity was represented by a system of differential equations (*Equations 1-3*) which was solved numerically using the Runge-Kutta method implemented in MATLAB (MathWorks) at a time step of 1 ms. Evaluations using smaller time steps (0.1 ms) were examined and produced similar results. At each time step, the model unit activities were updated based on their values at the previous step according to the differential equations. Considering the biological reality that spike rates cannot be negative, the activities were constrained to be non-negative. For the simulations including noise, we assumed an additive noise term for each unit, which evolved independently based on an Ornstein-Uhlenbeck process (*Equation 8*),

$$\tau_{noise} \frac{dNoise(t)}{dt} = -Noise(t) + \eta(t) \sqrt{\sigma^2 dt} \tag{8}$$

where $\sigma^2$ is the variance of the noise, $\eta$ is a Gaussian white noise with zero mean and unit variance, and $\tau_{noise}$ is the time constant for the noise fluctuation process. The time constant for the noise process ($\tau_{noise}$) was set to 2 ms, aligned with previous studies (*Wang, 2002*; *Wong and Wang, 2006*). Note that this approach assumes for convenience that noise arises in model unit activity; however, similar stochasticity can be implemented assuming noise arises in inputs external to the circuit, generalizing our findings.

All parameters used for visualization were set as the following unless specified elsewhere or fitted as free parameters in *Figure 6*: $\tau_R$, $\tau_G$, and $\tau_D$ were set as the same value of 100ms only for non-quantitative visualization purposes and fitted independently as free parameters in the model fittings;

the gain control weight $\omega_{ij}$ was set as a unit value of 1 for simplicity; the self-excitation weight $\alpha$ was set as 15; the disinhibition weight $\beta$ was assumed as zero in representation (i.e. $\beta_{off} = 0$) and set as 1.1 in WTA competition; the input values $V_1$ and $V_2$ were set as S*(1+c') and S*(1-c'), where c' indicates the motion coherence of the stimulus, with varied values (0, 3.2, 6.4, 12.8, 25.6, 38.4, and 51.2%), and S indicates the scale of input (set as 250). Baseline input $B_R$ was set as 70 in representation in **Figure 3**, fit as a free parameter in **Figure 4**, and set as 0 in WTA competition and persistent activity in **Figures 5–10**. Ornstein-Uhlenbeck noise was set as zero in most figures aiming at visualization of the model properties but $\sigma = 2$ in **Figure 10B–D** and fit as a free parameter in **Figure 6**. The set of parameters in **Figure 7** were adjusted to predict the multi-alternative choice data: $\alpha$ was set as 0; $\beta$ was set as 1.5; scaling parameter was set as 640 for both pre-motion and motion period but set as 427 for the first 190 ms of motion period to replicate the initial dip; all parameters were kept the same between two- and four-alternative choices. Parameters in **Figure 8** were adjusted between two- and five-item cases in order to get comparable scale of activities in visualization: for two-item case, S = 250, α = 15, and $\beta_{on}$ = .4; for five-item case, S = 50, α = 37.5, and $\beta_{on}$ = .1.

## Fitting the LDDM and the DNM to the neural firing rates of normalized value coding

In order to quantify the performance of the LDDM in fitting to the neural dynamics of normalized value coding and compare with the original DNM, we fit the equilibrium values of the LDDM and DNM to the dataset of normalized value coding (Figure 4 in **Louie et al., 2011**). In this task, monkeys are asked to represent the reward targets (1, 2, or 3) on the corresponding location of the screen. The neural activity in the response field receiving direct input $V_1$ is recorded. Different combinations of $V_1$, $V_2$, and $V_3$ are provided to the monkeys based on the associated volume of water in the presented targets (varying from 50, 100, 200, and 250 μl or omitted targets marked as 0), resulting in 28 data points.

To fit the DNM, we employed the following differential equations,

$$\begin{cases} \tau_R \frac{dR_i}{dt} = -R_i + \frac{V_i + B_R}{1 + G_i}; \\ \tau_G \frac{dG_i}{dt} = -G_i + \sum_j \omega R_j. \end{cases} \tag{9}$$

To fit LDDM, we employed **Equations 1-3**.

The direct input value ($V_i$) to each pool takes the value of the volume of water reward (μl) plus a baseline input value $B_R$. $\omega$ was set as 1. In the LDDM, there are additional terms of self-excitation weighted by $\alpha$, baseline gain control input $B_G$ fed into $G_i$, and coupling between $R_i$ and the disinhibitory neurons $D_i$ weighted by $\beta$.

To fit the predicted activities to the empirical mean firing rates during the sustain phase, we fit the predicted activities during the equilibria of these models. The equilibria of the two models were solved in **Equation 10** and **Equation 11**, respectively by taking the differential equations (**Equation 9** and **Equations 1-3**) to zero.

For DNM,

$$R_i = \frac{V_i + B_R}{1 + \sum_j R_j} \tag{10}$$

For LDDM,

$$R_i = \frac{V_i + B_R + \alpha R_i}{1 + B_G + \sum_j R_j - \beta R_i} \tag{11}$$

To fit the empirical activities with normalized scale, we need another scaling parameter $R_{max}$ to capture the arbitrary rescaling, which results in the following equations (**Equation 12** and **Equation 13**).

For DNM,

$$R_i = R_{max} \frac{V_i + B_R}{1 + \sum_j R_j} \tag{12}$$

For LDDM,

$$R_i = R_{max} \frac{V_i + B_R + \alpha R_i}{1 + B_G + \sum_j R_j - \beta R_i} \tag{13}$$

Since we assume the disinhibition modules in LDDM keep silent during representation, $\beta$ takes zero. For a trinary input system, the equilibria of $R_i$ the two models can be described by the following equations (**Equation 14** for DNM and **Equation 15** for LDDM).

For DNM,

$$\begin{cases} R_1 + R_2 + R_3 - \frac{R_{max}(V_1 + B_R)}{R_1} = 1; \\ R_2 + R_1 + R_3 - \frac{R_{max}(V_2 + B_R)}{R_2} = 1; \\ R_3 + R_1 + R_2 - \frac{R_{max}(V_3 + B_R)}{R_3} = 1. \end{cases} \tag{14}$$

For LDDM,

$$\begin{cases} R_1 + R_2 + R_3 - \frac{R_{max}(V_1 + B_R)}{R_1} = 1 + B_G - \alpha; \\ R_2 + R_1 + R_3 - \frac{R_{max}(V_2 + B_R)}{R_2} = 1 + B_G - \alpha; \\ R_3 + R_1 + R_2 - \frac{R_{max}(V_3 + B_R)}{R_3} = 1 + B_G - \alpha. \end{cases} \tag{15}$$

From **Equation 15**, we realized that $\alpha$ and $B_G$ share the same term and cannot be independently identified. Thus, we combined these parameters as one in our model fitting.

Based on the above analyses, two free parameters were estimated for the DNM (baseline input $B$ and the scaling parameter $R_{max}$). Three free parameters were estimated for the LDDM ($B_R$, $S$, and a combined parameter $B_G - \alpha$). The Bayesian adaptive direct search (BADS) algorithm (**Acerbi and Ma, 2017a**; **Acerbi and Ma, 2017b**) was implemented to minimize the ordinary squared error between the steady state of the predicted neural firing rates on $R_1$ and the empirical data.

## Fitting the RNM to the neural firing rates of normalized value coding

In order to quantify the performance of the RNM in predicting normalized value coding, we fit the reduced form of the RNM (**Wong and Wang, 2006**) with four free parameters ($JN_{i,i,i}$, $JN_{i,j,k(i \neq j \neq k)}$, $I_0$, and a scaling parameter $S$ applied to the predicted neural firing rates) to a normalized value coding dataset (Figure 4 in **Louie et al., 2011**). Other parameters are set the same as reported in the original paper (**Wong and Wang, 2006**), except that the noise term $\sigma$ is set as zero. The RNM is expanded to a trinary choice circuit, with three selective populations wired together based on the same rules specified in the original paper (**Wong and Wang, 2006**). We study the predicted neural activity on Pool 1 that receives direct input from $V_1$ and investigate how the activity of Pool 1 changes with the values of contextual inputs $V_2$ and $V_3$. The BADS algorithm was used to minimize the mean squared error between the predicted neural firing rates of Pool 1 and the empirical neural firing rates data reported in Figure 4 of **Louie et al., 2011**. The best-fitting result shows that the RNM explains 89.2% of the variance, worse than the DNM and LDDM we reported in the main text (best-fitting parameters: $JN_{i,i,i}$ = 0.0055, $JN_{i,j,k(i \neq j, i \neq k)}$ = 0.0861, $I_0$ = 0.3511, and $S$ = 1.074).

## Fitting the LDDM to empirical behavioral data

The LDDM with seven free parameters (the weights of self-excitation [$\alpha$] and disinhibition [$\beta$], the variance of Gaussian white noise in the Ornstein-Uhlenbeck process [$\sigma^2$], the scaling parameter of input [$S$], and time constants for three types of units $\tau_R$, $\tau_G$, and $\tau_D$) was fit to choice behavior (RT and choice accuracy) in a classic perceptual decision-making dataset (**Roitman and Shadlen, 2002**). $B_R$ was set to zero since any positive values will worsen the accuracy performance. $B_G$ was fixed as zero since it shows high collinearity with $\alpha$ (**Figure 6—figure supplement 3**). We employed the commonly used QMLE method (**Heathcote et al., 2002**; **Ratcliff and McKoon, 2008**). The rationale of QMLE is to minimize the differences between the predicted data and the empirical data on the proportion of trials located in each RT bin. Choice accuracy was implicitly estimated because the algorithm accounts for the proportion of trials between correct and error trials. Nine quantiles (from 0.1 to 0.9 with 0.1 of step size) were used, resulting in 10 RT bins, with correct and error trials accounted for separately at each coherence level. Because the LDDM has no closed-form analytic expression for the RT distribution, we evaluated the prediction by Monte Carlo simulations (10,240 repetitions for each input

coherence). In each simulated trial, the initial $R_1$ and $R_2$ activities were set as 32 Hz to be comparable to the empirical data (**Churchland et al., 2008**; **Roitman and Shadlen, 2002**). Visual stimulus (motion) inputs were defined as $S*(1+c')$ and $S*(1-c')$ for $V_1$ and $V_2$, where the free parameter $S$ models input scaling and the coherence $c'$ replicated values in the original experiment (0, 3.2, 6.4, 12.8, 25.6, and 51.2 %)(**Roitman and Shadlen, 2002**). A gap period (90ms) was implemented at visual stimulus onset as a non-decision period to capture the commonly observed initial dip untuned to inputs in empirical firing rates (**Roitman and Shadlen, 2002**). Gated disinhibition was activated along with inputs after the gap. A decision was reached when either of the $R$ unit activities reached a decision threshold of 70 Hz, the biological threshold observed in the empirical data (**Roitman and Shadlen, 2002**). 30 ms of non-decision time was added to the RT of threshold hitting to capture the delay in the down-streaming motor execution. After the decision, the input values, $\alpha$ and $\beta$, were reset to zero. The negative loglikelihood (nLL) of QMLE was minimized using BADS algorithm in MATLAB (**Acerbi and Ma, 2017b**). The estimation was conducted using GPU (NVIDIA Tesla V100) parallel computation on a high-performance cluster (NYU Langone), with 160 sets of random initial parameter values to prevent local minima. The set with the smallest nLL in its fitting result was selected as the best-fitting result.

The visualization of the predicted RT distribution (**Figure 6A**) was calculated based on 60 evenly distributed RT bins, with correct and error trials calculated separately under each coherence. The predicted neural dynamics (**Figure 6D**) were generated using the model best fit to behavior. $R$ unit activities were aggregated across correct trials, segregated by units associated with the chosen and unchosen sides. As in the original experiment data visualization (**Roitman and Shadlen, 2002**), activity early in trials was aligned to stimulus onset. Data within 100 ms of boundary crossing were omitted to reduce the impact of decision dynamics on visualizing early-stage ramping dynamics. Early activity traces were cut off at the median value of RT for each coherence level to ensure that the average trace was based on at least half of the trials. Activity late in trials was aligned to the time of the decision, and data within 200ms of stimulus onset was omitted.

## Fitting the RNM to empirical behavioral data

In order to compare the model performance in predicting choice behaviors, we fit the original RNM to the classical perceptual decision dataset (**Roitman and Shadlen, 2002**). We used the reduced form of the RNM (**Wong and Wang, 2006**). We set eight parameters in the reduced model (see the Appendix in its original paper) as free parameters to fit: self-excitatory coupling weights $JN_{1,1} = JN_{2,2}$, mutual inhibitory coupling weights $JN_{1,2} = JN_{2,1}$, non-selective input $I_0$, noise amplitude of Ornstein-Uhlenbeck (OU) process $\sigma_{noise}$, input scale $\mu_0$, synaptic kinetic parameter $\gamma$, initial value $H_0$, and time constant $\tau_S$. The other parameters that describe the input-output relationship of a single cell were set as the same in the paper: $a$=270 (VnC)$^{-1}$, $b$=108 Hz, and $d$=0.154 s. The time constant for the AMPA receptor $\tau_{AMPA}$ was fixed as 2 ms. The task setting, non-decision time (90 ms delay after stimulus onset and 30 ms delay before saccade), and optimization were kept the same as in fitting the LDDM (see above). The time step $dt$ was set as .001 s.

## Fitting the LCA to empirical behavioral data

Another widely acknowledged decision circuit model – the LCA model (**Usher and McClelland, 2001**) was fit to the behavioral data (**Roitman and Shadlen, 2002**). The dynamics of the two nodes in the LCA can be described using the following differential equations (**Equation 16**).

$$dx_i = \left( \rho_i - kx_i - \beta \sum_{j \neq i} x_j \right) \frac{dt}{\tau} + \xi_i \sqrt{\frac{dt}{\tau}} \qquad (16)$$

where $x_i$ ($i = 1$ and 2) indicates the activity of each node; $\rho_i$ indicates the excitatory input value to each node; $k$ indicates the net leakage on each node after the cancellation of recurrent excitation; $\beta$ weighs the mutual inhibition strength from the other nodes; $\xi_i$ is a Gaussian random noise on each node with an SD of $\sigma$.

The input values $\rho_i$ were set as $1+c'$ for Option 1 and $1-c'$ for Option 2, with $c'$ changing over 0 to 0.512. We fitted the threshold as a free parameter. In that way, the time constant $\tau$ can be taken as an arbitrary value (100 ms used in our case) since it was not independent from the threshold. Other than the parameters we mentioned above, non-decision time $T_0$ was fixed as 120 ms, sharing the same assumption with the other two models based on the empirically observed delays after stimulus onset (90 ms) and before the saccade (30 ms). That gives in total four free parameters to estimate

$k$, $\beta$, $\sigma$, and *threshold*. Since the scale of the activities is arbitrarily defined, it would need rescaling when compared to the empirical data of mean firing rates in the unit of Hz. The task setting and the optimization used were kept the same as in fitting the LDDM (see above). The time step $dt$ was set as 0.001 s.

## Analysis of persistent activity

We showed in **Results** that the LDDM with recurrent excitation predicts persistent activity that maintains input information during delay intervals. Here, we provide mathematical analyses of the LDDM differential equations to examine the properties and genesis of this persistent activity. In addition to examining the property of the system with symmetric gain-control weights ($\omega_{ii} = \omega_{ij(i \neq j)}$), we expanded our analysis to allow the gain-control weights to be asymmetric; this allows us to examine the robustness of LDDM properties to asymmetric weights.

Equilibrium states of the differential equations (**Equations 1-3**) after the withdrawal of inputs were considered. The gain control weights $\omega_{ij}$ were split into two parts, with the local-option weight denoted as $w$ ($\omega_{ii} = w$) and the cross-option weight denoted as $v$ ($\omega_{ij(i \neq j)} = v$). The input values were set to zero, and local disinhibition was assumed inactive ($\beta = 0$). Equilibria of the system were solved by taking the intersection of the steady states of all units, i.e., when $dR_i/dt$, $dG_i/dt$, and $dD_i/dt$ all equal to 0. When the input terms are set to zero, the solution degrades from **Equation 5** to **Equation 17** as a linear form,

$$wR_i^* + v\sum_{i \neq j} R_j^* = \alpha - 1 - B_G \tag{17}$$

For a binary choice system, the solution of **Equation 17** is denoted in linear algebra as:

$$\begin{bmatrix} w & v \\ v & w \end{bmatrix} \begin{bmatrix} R_1^* \\ R_2^* \end{bmatrix} = \begin{bmatrix} \alpha - 1 - B_G \\ \alpha - 1 - B_G \end{bmatrix} \tag{18}$$

The solutions of the equations depend on the value of recurrent excitation $\alpha$ and baseline gain control $B_G$. When $\alpha \leq 1 + B_G$, the equations do not provide a positive solution. This explains why the system without recurrent excitation ($\alpha = 0$) cannot generate persistent activity. When $\alpha > 1 + B_G$, the equations provide positive solutions. The model generates persistent activities in three different patterns depending on the symmetry of gain control weights, i.e., $v < w$, $v = w$, and $v > w$.

First, by assuming $v = w$ and $\alpha > 1 + B_G$, the nullclines of $R_1$ and $R_2$ overlap on a line of attraction, as shown in **Figure 8B** (the same as **Figure 8—figure supplement 1B**). Any position on this line is an equilibrium point. This is a special case where the eigenvalues on each point have a real part of zero; therefore, linearization around the equilibrium points cannot tell us their stability. Thus, we checked the instantaneous change direction of neural activities instead across a wide range of initial values to see whether the system converges to the line of attraction. From the differential equations (**Equations 1-3**), the ratio $dR_1/dR_2$ of the instantaneous change rates of $R_1$ ($dR_1/dt = R_1\left(1 - \frac{\alpha}{1+G_1}\right)$) and $R_2$ ($dR_2/dt = R_2\left(1 - \frac{\alpha}{1+G_2}\right)$) keeps the same ratio as the ratio of original activities ($R_1/R_2$), given $G_1 = G_2$ under the assumption of symmetric gain control weights. As a result, for any given initial values, $R_1$ and $R_2$ activities change in the direction that preserves the original ratio until reaching equilibrium on the line of attraction. The instantaneous changes of $R_1$ and $R_2$ are shown as a vector field (red arrows) in **Figure 8B**. Thus, any positive initial values will drop into an equilibrium state with the ratio of $R_1^*/R_2^*$ maintaining the ratio of initial values, which preserves the ratio of inputs when the activities are inherited from the stage of value representation. **Figure 8—figure supplement 1E** shows example dynamics of $R_1$ and $R_2$ under different ratios of input values (**Figure 8—figure supplement 1G**). The activities show the characteristic dynamics of divisive normalization during the inputs and preserve this input information after the withdrawal of inputs.

However, since the values of $R_1^*$ and $R_2^*$ are complementary on the line of attraction, any combination of values with a constant sum satisfies the equilibrium. Thus, any disturbance to the system (e.g. random noise) will drive $R_1^*$ and $R_2^*$ to deviate from their original ratio resulting in a loss of the coded information about the inputs. Noise-driven drift on the line of attraction will cause the decaying of the coded value information over time, consistent with the degradation attribute of working memory

(*Barrouillet et al., 2011*; *Barrouillet and Camos, 2012*; *Lee and Harris, 1996*; *Paivio and Bleasdale, 1974*; *Portrat et al., 2008*).

In addition, under the special condition of symmetric gain control weights ($v = w$), the formula in *Equation 18* can be easily expanded to multiple inputs with the equilibrium delay interval activities defined by:

$$\sum_i^N R_i^* = \frac{\alpha - 1 - B_G}{w} \tag{19}$$

The summed value of all $R$ units equals to a constant $\frac{\alpha - 1 - B_G}{w}$. When the number of inputs ($N$) increases, the activity shared by each $R$ unit decreases and leads to a lower signal relative to the noise scale. Thus, as the number of coded items increases, the information kept during persistent activity may become less accurate considering a lower signal-to-noise ratio. This may explain another important attribute of working memory – the constraint of the working memory span (*Cowan, 2010*; *Cowan, 2016*; *Engle, 2001*; *Engle, 2002*; *Oberauer et al., 2016*).

Second, by assuming $v < w$ and $\alpha > 1 + B_G$, the nullclines of $R_1$ and $R_2$ intersect on a unique equilibrium point, where $R_1$ and $R_2$ share the same value $\frac{\alpha - 1 - B_G}{w + v}$ (*Figure 8—figure supplement 1A*). The point is confirmed as attractive by linearization. Any positive initial values on the space of $R_1$ and $R_2$ will converge into this point, which is visualized in the instantaneous change ranges of $R_1$ and $R_2$ (red arrows) for a wide range of given initial values (*Figure 8—figure supplement 1A*). Thus, $R_1$ and $R_2$ will gradually converge to be equal, and the original information about input values will be lost. Nevertheless, the dynamic of information loss is based on the level of asymmetry of $\omega_{ij}$. For a close-to-symmetric $\omega_{ij}$ matrix, the input information can still be preserved for a considerable amount of time. We showed example dynamics of information loss in *Figure 8—figure supplement 1D*. After the withdrawal of inputs, the $R$ unit activities collapse into the same level and the coded ratio information gradually diminishes (simulation parameters: $\alpha = 10$, $B_G = 0$, $w = 1$, $v = .7$, $\beta = 0$).

Finally, by assuming $v > w$ and $\alpha > 1 + B_G$, the nullclines of $R_1$ and $R_2$ intersect on a unique equilibrium point, which is confirmed as unstable by linearization (*Figure 8—figure supplement 1C*). Any initial values of activities on the space will diverge into the upper-left or bottom-right corner of the space generating high contrast between $R_1$ and $R_2$, with the higher activity as $\frac{\alpha - 1 - B_G}{w}$ and the lower activity suppressed to zero. The instantaneous change rates of $R_1$ and $R_2$ (red arrows) are visualized in the vector field in *Figure 8—figure supplement 1C*. The instantaneous change direction bifurcates at the line of $R_1 = R_2$, biased to the side associated with the higher initial activity. As an outcome, the $R$ unit with higher initial values tends to increase while the opponent unit tends to be suppressed to zero, a process that implements WTA competition before the action stage but with constrained higher activity. Example $R_1$ and $R_2$ activity dynamics are shown in *Figure 8—figure supplement 1F*. After the withdrawal of inputs, $R_1$ activities with different preceding input values collapse onto the same level of high activity, while $R_2$ activities with lower input values are suppressed to zero. Thus, the system gradually switches from the normalized coding of inputs to a categorical coding of choice over the delay interval.

We also examined whether persistent activity could exist with active local disinhibition. We showed in **Results** that persistent activity in the working-memory task switches to WTA choice under the dynamic control of disinhibition (*Figure 8D–F*). How does the transition from persistent activity to WTA choice happen? How might disinhibition change the dynamic pattern of persistent activity during a delay interval?

The analysis was based on the differential equations of the system with symmetric gain control weights and without inputs (*Equations 1-3*). The equilibrium solution is given by:

$$(\omega - \beta) R_i^* + \omega \sum_{i \neq j} R_j^* = \alpha - 1 - B_G \tag{20}$$

With binary inputs, the solution can be thus written as:

$$\begin{bmatrix} \omega - \beta & \omega \\ \omega & \omega - \beta \end{bmatrix} \begin{bmatrix} R_1^* \\ R_2^* \end{bmatrix} = \begin{bmatrix} \alpha - 1 - B_G \\ \alpha - 1 - B_G \end{bmatrix} \tag{21}$$

Besides the impact of recurrent excitation and baseline gain control discussed above, equilibrium responses are determined by the relative strength between disinhibition ($\beta$) and the gain control

weight ($\omega$). We examined three separate conditions: $\beta = 0$, $0 < \beta < \omega$, and $\beta > \omega$. We have already shown the analysis for the special case when $\beta = 0$ above (phase plane analysis and example dynamic shown in *Figure 8—figure supplement 1B*) and replotted in *Figure 8—figure supplement 2A* for the sake of comparison with the other two conditions.

By assuming $0 < \beta < \omega$, the nullclines of $R_1$ and $R_2$ intersect on a unique equilibrium point, whose stability was confirmed as unstable after checking the eigenvalues of the Jacobian matrix around the point (*Figure 8—figure supplement 2B*). Any initial values on the space will diverge into the upper-left or bottom-right corner of the space, with the higher activity value as $\frac{\alpha-1-B_G}{\omega-\beta}$ and the lower activity value as zero. We show the instantaneous change rates of $R_1$ and $R_2$ at given initial values in the vector field (red arrows; *Figure 8—figure supplement 2B*). In *Figure 8—figure supplement 2E*, we show the examples $R_1$ and $R_2$ activity dynamics (value setting kept the same as in *Figure 8—figure supplement 1G*). All of the $R_1$ with larger input values converge into the same level of activity after the withdrawal of inputs, while all of the $R_2$ with lower input values are suppressed to zero, implementing a WTA competition. Thus, the system gradually switches from normalized coding of input values to categorical choice from the early to the late stage of persistent activity.

By assuming $\beta > \omega$, most of the features are similar to the previous situation, except that the model now predicts no constraints on the maximum activity (*Figure 8—figure supplement 2C*). The system shows nullclines with an intersection at a unique repellor. The activities of $R_1$ and $R_2$ bifurcate at the line of $R_1 = R_2$. The example dynamics show that the activity of $R_1$, which has higher initial value, increases to an unlimited level and thus will reach a decision threshold. The rising speed of $R_1$ depends on the advantage of $R_1$ over $R_2$ as defined by their initial values.

Taken together, these analyses show that persistent activity is present as normalized coding of input values only with symmetric gain control weights ($w = v$) and inactive disinhibition ($\beta$). When disinhibition has a moderate strength ($0 < \beta < \omega$), the persistent activity gradually transitions from value coding to categorical choice coding but avoids hitting the decision threshold. When disinhibition is strong enough ($\beta > \omega$), the system generates WTA competition and reaches the decision threshold.

## Simulation of pharmacological manipulation of inhibitory activity

In *Figure 10*, we tested inhibitory potentiation (e.g. GABAergic agonist) manipulation effects in both the LDDM and RNM (*Wong and Wang, 2006*) by assuming different levels of enhancement of the inhibitory projections. For LDDM (*Figure 10A–D*), we assumed $\omega_{ij} = 1$, $\tau_R = \tau_G = \tau_D = 100\ ms$, input scale $S$=256, decision threshold =70 Hz, and dt =1 ms. Panel **A** illustrated the temporal dynamic of excitatory pools ($R_1$ and $R_2$) under input coherence of 25% between control (inhibitory connection weight = 1.0) and potentiation (inhibitory connection weight =3.8) conditions (other parameters used were $\alpha = 5$, $B_R = 0$, $B_G = 0$, $\beta = 1.4$, and $\sigma = 0$). Panel **B** examined the predicted RT and choice accuracy over different input coherences (c' = [0, 3.2, 6.4, 12.8, 25.6, and 51.2%]) and levels of inhibitory weights (from 1 [control] to 4 [enhanced]; $\alpha = 10$, $B_R = 0$, $B_G = 0$, $\beta = 1.1$, $\sigma = 2$, and 10,000 repetitions). Panel **C** showed the chromomeric and psychometric curves over a number of input coherences (1–100%) under the section between control and inhibitory potentiation (inhibitory connection weight = 1.8). Panel **D** scanned the full parameter space of $\alpha$ and $\beta$ between the contrast of control and inhibitory potentiation (inhibitory connection weight = 1.8; c'=3.2%, $B_R = 0$, $B_G = 0$, $\sigma = 2.0$, and 10,000 repetitions). For RNM (*Figure 10E–G*), we used the parameters specified in *Wong and Wang, 2006* for the mean-field rate model. Inhibitory potentiation was manipulated by weighting the inhibitory connection in the model. Panel **E** illustrated the noiseless neural dynamics of RNM using the same input coherences and inhibitory enhancement levels as in panel **A**. Panel **F** was set to compare with panel **B**; thus the input coherences and inhibitory enhancement kept the same as in panel **B**, with noise amplitude set as $\sigma = .02$ recommended by *Wong and Wang, 2006*. Panel **G** showed the chromomeric and psychometric function predicted by RNM under the same input and inhibitory potentiation assumptions as in panel **C**.

## Motifs tested and compared for normalized coding and WTA choice

We tested a series of motifs and found that local disinhibition is critical for integrating normalized valuation and choice functions. To do this, we tested four types of modifications that might enhance mutual competition between the option-specific local sub-circuits (*Figure 2—figure supplement 1A*): (a) *Recurrent self-excitation* (loops weighted by $\alpha$), with self-amplification of each $R$ unit, a property

shown to be important for mutual competition in the RNM. (b) *Local disinhibition* (loops weighted by $\beta$), which is the focus of the main text, mediated through disinhibitory units (*D*); the function of a *D* unit is to inhibit the gain control *G* unit in the local sub-circuit, therefore, release inhibition on the local *R* units. (c) *Cross inhibition* (loops weighted by $\eta$), which directly inhibits the lateral *R* units through inhibitory units (*I*) to implement mutual inhibition. (d) *Lateral gain control boost* (loops weighted by $\gamma$), which is mediated through excitatory units (*E*) to boost the lateral *G*, therefore, drives higher gain control on the lateral *R* than the local *R* (i.e. asymmetric gain control) and realizes mutual inhibition.

To see which type of modification(s) is/are critical for integrated value normalization and choice, we tested different combinations of these modifications on the original DNM circuit. The full model with all modifications can be described by a set of differential equations (*Equations 22-26*):

$$\tau_R \frac{dR_i}{dt} = -R_i + \frac{V_i + \alpha R_i - I_j}{1 + G_i} \tag{22}$$

$$\tau_G \frac{dG_i}{dt} = -G_i + \omega \sum_{j=1}^{N} R_j + E_j - D_i \tag{23}$$

$$\tau_D \frac{dD_i}{dt} = -D_i + \beta R_i \tag{24}$$

$$\tau_I \frac{dI_i}{dt} = -I_i + \eta R_i \tag{25}$$

$$\tau_E \frac{dE_i}{dt} = -E_i + \gamma R_i j \tag{26}$$

where *i*=1, ..., *N* designates choice alternatives, each of which receives input $V_i$, and $\tau_R$, $\tau_G$, $\tau_D$, $\tau_I$, and $\tau_E$ are the time constants for the *R*, *G*, *D*, *I*, and *E* units. The weights $\omega$ represent the coupling strength between excitatory units *R* and gain control units *G*. The parameters $\alpha$, $\beta$, $\eta$, and $\gamma$ control the active state of recurrent excitation, local disinhibition, cross inhibition, and lateral gain control boost loops, respectively.

The active and inactive states of the four types of loops can be combined into $2^4$=16 possiblegf models. Example dynamics were shown in *Figure 2—figure supplement 1B* for each type of model. When local disinhibition ($\beta$) is off (left two columns), the model generates WTA dynamics only when cross inhibition ($\eta$) is on. However, the maximum activity in the late stage is still restricted to a value lower than the phasic peak during the early stage, contradicting empirical findings that the late-stage decision threshold is usually higher than activity in the early phasic peak (*Churchland et al., 2008*; *Kiani et al., 2008*; *Kiani and Shadlen, 2009*; *Louie et al., 2011*; *Roitman and Shadlen, 2002*; *Rorie et al., 2010*; *Shadlen and Newsome, 2001*; *Sugrue et al., 2004*). This restriction arises because, with only cross inhibition, local option gain control is not released; this release requires local disinhibition. With local disinhibition on ($\beta > 0$, the right two columns), the models generate WTA dynamics with high activity in the late stage to reach the decision threshold. This is robust even without any other modifications (see the panel with $\eta$ and $\gamma$ off), highlighting the role of local disinhibition in generating WTA competition. For the sake of simplicity, we omitted other non-essential modifications and kept only the loop of local disinhibition. Because recurrent excitation is important for persistent activity and exists widely in cortical circuits, we retained it as well. The modified DNM model with local disinhibition and recurrent self-excitation is the primary model (LDDM) characterized in the current work.

## Additional information

### Funding

| Funder | Grant reference number | Author |
| --- | --- | --- |
| National Institutes of Health | R01DA038063 | Paul Glimcher |
| National Institutes of Health | R01DA043676 | Paul Glimcher |

The funders had no role in study design, data collection and interpretation, or the decision to submit the work for publication.

## Author contributions
Bo Shen, Conceptualization, Data curation, Software, Formal analysis, Validation, Investigation, Visualization, Methodology, Writing – original draft, Writing – review and editing; Kenway Louie, Conceptualization, Supervision, Funding acquisition, Methodology, Writing – original draft, Writing – review and editing; Paul Glimcher, Conceptualization, Resources, Supervision, Funding acquisition, Writing – original draft, Project administration, Writing – review and editing

## Author ORCIDs
Bo Shen ![ORCID] http://orcid.org/0000-0001-5796-0844
Kenway Louie ![ORCID] http://orcid.org/0000-0002-9665-5436
Paul Glimcher ![ORCID] http://orcid.org/0000-0001-7872-3856

## Decision letter and Author response
Decision letter https://doi.org/10.7554/eLife.82426.sa1
Author response https://doi.org/10.7554/eLife.82426.sa2

---

# Additional files

## Supplementary files
• MDAR checklist

## Data availability
The empirical data presented in this paper and MATLAB code used for simulations and fitting the empirical data is available at DOI https://doi.org/10.17605/OSF.IO/YGR57.

The following dataset was generated:

| Author(s) | Year | Dataset title | Dataset URL | Database and Identifier |
|---|---|---|---|---|
| Shen B, Louie K, Glimcher P | 2023 | Flexible control of representational dynamics in a disinhibition-based model of decision making | https://doi.org/10.17605/OSF.IO/YGR57 | Open Science Framework, 10.17605/OSF.IO/YGR57 |

The following previously published dataset was used:

| Author(s) | Year | Dataset title | Dataset URL | Database and Identifier |
|---|---|---|---|---|
| Roitman JD, Shadlen MN | 2002 | Response of neurons in the lateral intraparietal area during a combined visual discrimination reaction time task | https://shadlenlab.columbia.edu/resources/RoitmanDataCode.html | Matlab m files, shadlenlab |

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
