## [Editor Report]

This novel theoretical work outlines a unifying architecture for decision-making via disinhibition. The model clearly links observations across multiple empirical studies and highlights how characteristics from previous decision models can be effectively integrated into a single mechanism. This will be of interest to a wide variety of neuroscientists who work across levels of analysis.

---

## [Decision Letter]

**Decision letter after peer review:**

Thank you for submitting your article "Flexible control of representational dynamics in a disinhibition-based model of decision making" for consideration by *eLife*. Your article has been reviewed by 3 peer reviewers, including Timothy Verstynen as the Reviewing Editor and Reviewer #1, and the evaluation has been overseen by Michael Frank as the Senior Editor. The following individual involved in the review of your submission has agreed to reveal their identity: Alexandre Filipowicz (Reviewer #2).

Essential revisions:

Based on the combined reviews and discussions among the reviewers and review editor, the following issues should be addressed as essential revisions for moving forward.

1) Competing models: While all three reviewers agree that a network that combines both value-based and WTA dynamics is interesting and useful, there is consensus that the lack of competing or contrastive models (beyond the component models that are combined to make the LDDM) tempers the conclusions that can be taken away from the work. The LDDM should be compared with reasonable competing models and, conceptually, the authors should highlight what the LDDM adds over existing models).

2) Parameter specificity: There is a consensus in the reviews regarding questions about the necessity, specificity, and interpretation of some of the model parameters. Reviewer 3, in particular, points out potential inconsistencies in the way the disinhibition Β parameter is interpreted. Reviewer 2 highlights confusion between the necessary and specific role of Α compared to Β in some of their simulations. As both reviewers point out, an examination of the optimization surface of the fits (Reviewer 3) and/or a parameter recovery analysis (Reviewer 2) could help demonstrate the robustness and identifiability of the LDDM model parameters. This would also address Reviewer 1's concern about model complexity and overfitting.

3) Conceptual framing: The reviews point out that the conceptual framing of the goals shifts across the different sections of the manuscript. The authors should be clear about the high-level framing (Reviewer 1), whether it's about the integration of two decision frameworks or the role of disinhibition). The author should also clarify the precise interpretation of the key aspects of the model that explain its behavior (Reviewers 1 and 3). Finally, the authors should more extensively link their model in the context of prior work (Reviewers 1 and 3)

*Reviewer #1 (Recommendations for the authors):*

1. I recommend the authors take time to more explicitly clarify the goal of the study. What is the singular take-home message that the reader should take away from this? This singular message should be tempered enough so as not to overstate this as the first unification of value normalization and response selection, but more specific to what is being tested.

2. I recommend adding a discussion on known disinhibition circuits like the cortical-basal ganglia loops and showing how the LDDM links to prior models of these networks.

3. I would recommend finding a non-DNM and non-RNM control model to compare the LDDM against.

4. I recommend using model fit metrics to evaluate how well the LDDM (and a control model) explain the neurophysiological data.

*Reviewer #3 (Recommendations for the authors):*

1) I think the authors will have to rewrite parts of the manuscript to address the concerns I raise – (i) especially to clarify the precise interpretation of the single key parameter that determines the behavior of the model, and (ii) point out the connection to previous work (Machens et al. 2005, Yang et al., 2016, Litwin-Kumar et al. 2016) emphasizing the specific ways in which their work is an advance on these previous studies.

2) It would also greatly help if the usage of notation is made consistent throughout the paper. For instance, in the figures and the equations in the main text (Equations 1-3), disinhibition is denoted as D, but in the methods (Equations 5-8) and the supplementary figures (Figure 2, Supplementary Figure 1) it is denoted as 'I'.

3) I appreciate that the authors also studied a more general and 'extended' version of their model (of which the LDDM is a special case) and explore how it behaves in different regions of parameter space (Figure 2, Supplementary Figure 1). However, I found the general description of their extended model quite confusing, particularly, some of the design choices. For instance, the extended model consists of additional excitatory units (E) that are referred to as 'gain control boost loops'. These are never mentioned in the main text and their purpose for the overall story of the paper seems somewhat unclear to me. Since the R units already have projections to both 'local' and 'lateral' gain units (through 'omega', Figure 2A), couldn't the E units simply be replaced by stronger self-recurrence on the R units?

4) The most interesting analyses in the paper are where the authors fit the circuit model to neurophysiological data. The authors then report the values of the fitted parameters and also perform the model comparisons by reporting AIC/likelihood ratios. However, if possible, it would be very informative to also visualize the optimization surface of these fits to understand whether some of the free parameters trade-offs against one another, as I think that would affect the overall conclusions drawn in the paper, and also convince me about the robustness of the fitted parameters.

5) A somewhat more open-ended question is about the choice of the time constants for the 3 types of units in the model (R, G, and D), which appear to be fixed to a value of 100ms for all the results presented in the manuscript. Can the authors justify this choice? Considering that SST (which, I presume are the gain control units G) and VIP neurons have fundamentally different conductance profiles and are known to show an entire range of spiking patterns (Tremblay et al., 2016), is it justifiable to assume that their time constants all have the same value?

6) In general, the presentation of the figures can be improved:

a) In Figure 2—figure supplement 1, I should be replaced by D. Also, the parameter γ seems to be missing from the rows in subpanel B of this figure.

b) In Figure 5, it's hard to follow which subpanels are the 'main' subpanels and which ones are the insets.

c) The legend of Figure 4B (right column) seems to have an extra set of dots (ones that indicate the legend of V_in)

---

## [Author Response]

Essential revisions:Based on the combined reviews and discussions among the reviewers and review editor, the following issues should be addressed as essential revisions for moving forward.1) Competing models: While all three reviewers agree that a network that combines both value-based and WTA dynamics is interesting and useful, there is consensus that the lack of competing or contrastive models (beyond the component models that are combined to make the LDDM) tempers the conclusions that can be taken away from the work. The LDDM should be compared with reasonable competing models and, conceptually, the authors should highlight what the LDDM adds over existing models).

In the revision, we now also compare LDDM performance with an additional standard circuit model of decision-making – the leaky competing accumulator model (LCA). Model comparison shows that the LDDM outperformed both the RNM and the LCA in fitting a standard decision-making dataset. We would like to point out that the RNM has been prevalent for two decades, and we believe it is appropriate to be considered as one of the standard decision-making circuit models. However, the LCA is a widely known alternative dynamical model of decision-making suitable for behavioral comparison. Thus, we have now added the LCA as model fits to monkey WTA behavior, allowing a comparison across the three models (LDDM, RNM, LCA); in addition, we also quantify and compare how the option-coding unit activity in each best-fit model matches neurophysiological data.

Based on the Reviewers’ suggestions, we also now highlight two fundamental differences between the LDDM and other existing circuit models. First, we now highlight that the common inhibitory motif in existing standard decision-making models (e.g., RNM) is non-selective inhibition. We clarify here in the revision that the LDDM has a fundamental difference from RNM in predicting selective (or structured) inhibition, which has been identified in recent empirical studies using advanced neural imaging techniques. Second, we now explain that disinhibition in the LDDM provides a circuit mechanism for a switch between valuation and WTA dynamics; such a mechanism does not exist in simpler models like the RNM. Thus, while we show that the LDDM can outperform alternative models quantitatively, we now highlight that the LDDM is a novel framework that is qualitatively different from previous models, can accommodate new empirical findings (e.g., selective inhibition), and predicts specific hypotheses (e.g., differential activity of different inhibitory interneuron subtypes) for testing in future studies.

2) Parameter specificity: There is a consensus in the reviews regarding questions about the necessity, specificity, and interpretation of some of the model parameters. Reviewer 3, in particular, points out potential inconsistencies in the way the disinhibition Β parameter is interpreted. Reviewer 2 highlights confusion between the necessary and specific role of Α compared to Β in some of their simulations. As both reviewers point out, an examination of the optimization surface of the fits (Reviewer 3) and/or a parameter recovery analysis (Reviewer 2) could help demonstrate the robustness and identifiability of the LDDM model parameters. This would also address Reviewer 1's concern about model complexity and overfitting.

We thank the Reviewers for pointing out inconsistencies in notation regarding the β parameter. In the revision, we have fixed the notation problem on the β parameter (sorry about that) such that it is consistent throughout the manuscript. The manuscript now clarifies that in the LDDM the β parameter controls *D* unit response to *R* unit activity, and we now highlight that we conceptualize it as a measure of the functional connectivity between *R* and *D* neurons.

Regarding the role of α and β, we clarify in the response below and in the revision the uniqueness of α and β contributions to the dynamics of the system. The main takeaway, now highlighted in the text, is that for WTA selection dynamics in the LDDM, β is required while α contributes but is not necessary. In the revision, we now further examine and visualize the optimization surface of the fits for all parameters (as requested). Most of the parameters show smooth likelihood spaces, a narrow optimization range, and small collinearities, which indicate good identification of the model parameters and parsimonious model complexity. The α parameter shows less precise identification and shares some collinearity with β. We examined this problem carefully and realized that the α and β parameters make differential contributions to the shape of reaction time distribution in perceptual choice; these relatively small but potentially important differences in the likelihood space is an interesting phenomenon that we plan on targeting in future studies. Overall, after careful examination on the model fitting including parameter recovery analyses, we find that the model fitting is reliable and the parameters settings are parsimonious. These new analyses are provided in a series of additional figures and additional text (see below for specifics).

3) Conceptual framing: The reviews point out that the conceptual framing of the goals shifts across the different sections of the manuscript. The authors should be clear about the high-level framing (Reviewer 1), whether it's about the integration of two decision frameworks or the role of disinhibition). The author should also clarify the precise interpretation of the key aspects of the model that explain its behavior (Reviewers 1 and 3). Finally, the authors should more extensively link their model in the context of prior work (Reviewers 1 and 3).

The issue of conceptual framing is an important one, and we thank the Reviewers for pointing out the changing framing in the original paper. We have now edited the paper to highlight the importance of inhibition in computational models of decision-making, focusing throughout on two issues in the existing literature: (1) the assumption of non-selective, pooled inhibition in decision models (which is not supported by recent empirical evidence), and (2) widespread empirical evidence for interneuron diversity and complexity in inhibition circuits. Given this framing, the paper is now framed by introducing disinhibition into a circuit model of decision-making. In this framing, the integration of normalized value coding and WTA activity remains essential as a test of the validity of the LDDM.

Regarding the interpretation of model features, we have revised the paper to emphasize the relative importance of the β and α parameters to LDDM behavior, the conceptual role of β as a functional coupling between excitatory and disinhibitory units, and the interpretability of fit α values in terms of anatomical recurrence. Finally, we have now also added significant new text to the Discussion to place the LDDM in the proper context of previous computational work. As suggested by Reviewer #1, this includes a discussion of models of action selection in cortical-basal ganglia loops, which also utilized disinhibition (though with some differences that the LDDM, this is now discussed explicitly). As suggested by Reviewer #3, we also discuss how the LDDM is related to other computational models that use disinhibitory circuit motifs and address the functional processes relevant to the LDDM (divisive normalization, working memory, and decision-making).

Reviewer #1 (Recommendations for the authors):1. I recommend the authors take time to more explicitly clarify the goal of the study. What is the singular take-home message that the reader should take away from this? This singular message should be tempered enough so as not to overstate this as the first unification of value normalization and response selection, but more specific to what is being tested.

Upon reflection, we agree with Reviewer #1 that the original manuscript was not sufficiently clear in its emphasis, touching on a number of features of the model and findings. Ultimately, we believe that the most important aspect of the LDDM is disinhibition, and the most salient contribution of this paper is showing the advantages of a decision-making circuit that incorporates disinhibition (along with known features such as recurrent excitation and lateral and/or divisive inhibition). Driven by advances in genetic and imaging technologies, recent experiments have examined the diversity of inhibitory neuron subtypes and interneuron-interneuron connectivity, revealing evidence for widespread circuits implementing network disinhibition and driving behavioral and cognitive functions. However, disinhibition is not a part of standard circuit models of decision-making, and the potential contributions of disinhibition to neurophysiological and behavioral aspects of decision-making are unknown.

Thus, the overarching goal of the paper is to design, implement, and characterize the behavior of a decision-making circuit with disinhibitory motifs. The unification of value normalization and WTA activity is important, but primarily as a demonstration of the validity of the disinhibition-based LDDM model. As demonstrated in the paper, there are additional important capabilities of the LDDM, including the generation of structured inhibition and the provision of a mechanism for top-down control of circuit dynamics and function.

To make the take-home message clearer, we have now substantially revised the Abstract to emphasize the goal of incorporating disinhibition into a dynamical circuit model of decision-making:

“Inhibition is crucial for brain function, regulating network activity by balancing excitation and implementing gain control. Recent evidence suggests that beyond simply inhibiting excitatory activity, inhibitory neurons can also shape circuit function through disinhibition. While disinhibitory circuit motifs have been implicated in cognitive processes including learning, attentional selection, and input gating, the role of disinhibition is largely unexplored in the study of decision-making. Here, we show that disinhibition provides a simple circuit motif for fast, dynamic control of network state and function. This dynamic control allows a novel disinhibition-based decision model to reproduce both value normalization and winner-take-all dynamics, the two central features of neurobiological decision-making captured in separate existing models with distinct circuit motifs. In addition, the disinhibition model exhibits flexible attractor dynamics consistent with different forms of persistent activity seen in working memory. Fitting the model to empirical data shows it captures well both the neurophysiological dynamics of value coding and psychometric choice behavior. Furthermore, the biological basis of disinhibition provides a simple mechanism for flexible top-down control of network states, enabling the circuit to capture diverse task-dependent neural dynamics. These results suggest a new biologically plausible mechanism for decision-making and emphasize the importance of local disinhibition in neural processing.”

In addition, we have restructured the Introduction text to frame the importance of incorporating disinhibition into computational decision models. The primary change is to the initial paragraphs, though we have streamlined the Introduction as well:

“Inhibition is an essential component in neural network models of decision-making. In standard decision models, pools of option-selective excitatory neurons compete in a winner-take-all selection process via feedback inhibition (Roach et al., 2023; X.-J. Wang, 2002; Wong and Wang, 2006). Generally, such inhibition is thought to be homogenous and non-selective, with a single pool of inhibitory neurons receiving broad excitation, and in turn inhibiting excitatory neurons. However, more recent empirical findings suggest that inhibitory neurons interact with the decision circuit in a more structured manner. Inhibitory neurons active in decision-making exhibit choice-selective activity on par with excitatory neurons in the frontal cortex (Allen et al., 2017), parietal cortex (Allen et al., 2017; Najafi et al., 2020), and striatum (Gage et al., 2010), in contrast to the non-selective or broadly tuned inhibition seen in visual cortex during stimulus representation (Bock et al., 2011; Chen et al., 2013; Hofer et al., 2011; Kerlin et al., 2010; Liu et al., 2009; Niell and Stryker, 2008; Sohya et al., 2007). At an anatomic level, inhibitory interneurons also exhibit a remarkable diversity in morphology, connectivity, and physiological functions (Kepecs and Fishell, 2014; Markram et al., 2004; Tremblay et al., 2016). A prominent circuit motif observed in these anatomical studies is local disinhibition, in which vasoactive intestinal peptide (VIP)-expressing interneurons inhibit the neighboring interneurons expressing somatostatin (SST) or parvalbumin (PV) that inhibit dendritic or perisomatic areas in pyramidal neurons, so that locally disinhibit the activities of the pyramidal neurons in the neighboring area (Chiu et al., 2013; Fino and Yuste, 2011; Fu et al., 2014; Karnani et al., 2014, 2016; S. Lee et al., 2013; Letzkus et al., 2011; Pfeffer et al., 2013; π et al., 2013; UrbanCiecko and Barth, 2016). Here we explore the computational implications of that motif in decision-making.

While disinhibitory circuit motifs have been implicated in cognitive processes including learning, attentional selection, and input gating (Fu et al., 2014; Letzkus et al., 2011; X.-J. Wang and Yang, 2018), how disinhibition functions in decision-making circuits is unknown. Local circuit inputs to the VIP neurons suggest that disinhibition may be a key mechanism for generating the mutual competition necessary for option selection in decision-making. In addition, given the existence of long-range inputs (Kepecs and Fishell, 2014; S. Lee et al., 2013; Pfeffer et al., 2013; π et al., 2013; Schuman et al., 2021) and neuromodulatory inputs (Alitto and Dan, 2013; Fu et al., 2014; Pfeffer et al., 2013; Prönneke et al., 2020; Rudy et al., 2011; Tremblay et al., 2016) to the VIP neurons, local disinhibition has been proposed to play a particular role in dynamic gating of circuit activity; such gating may be essential in decision circuits underlying flexible behavior, mediating top-down control of network function (Fu et al., 2014; Kamigaki, 2019; S. Lee et al., 2013; Letzkus et al., 2011; π et al., 2013; Schuman et al., 2021; S. Zhang et al., 2014). Here we hypothesize that disinhibition controls a transition between information processing states, allowing a single decision-making circuit to both represent the values of alternatives and select a single best option amongst those alternatives.”

We have also added additional framing sentences in the Discussion (p. 32, lines 684 – 689):

“The prevalence of disinhibitory circuit motifs in the brain, and recent evidence for structured decision-related inhibitory activity, argue for a more structured implementation of inhibition in computational models of decision-making than has been previously employed. Here, we show that the disinhibition-based LDDM replicates three characteristic features of observed neurobiological decision-making circuits – normalized value coding, WTA choice, and persistent activity – for the first time within a single circuit architecture.”

2. I recommend adding a discussion on known disinhibition circuits like the cortical-basal ganglia loops and showing how the LDDM links to prior models of these networks.

The reviewer rightly points out an existing and important body of work that examines disinhibition circuits functioning in motor selection (and perhaps other types of selection), specifically cortical-basal ganglia loops (CBG). Disinhibition has multiple roles in the pathways that comprise CBG circuitry, the most relevant being the direct pathway where cortical activation of striatal GABAergic medium spiny neurons inhibits tonically active GPi/SNr inhibitory neurons, thus releasing downstream neurons in the output pathway (thalamus). LDDM is related to precursor CBG models because, in both cases, activation of disinhibition is part of the selection process. This important historical link is now acknowledged in the manuscript.

A crucial difference between disinhibition in CBG and LDDM is the selectivity of the disinhibition and its role in specifying versus initiating choice. In standard models of the CBG, disinhibition is selective and favors the option to be chosen; this selective or biased disinhibition is driven by differences in input (in simple selection models) or differences in synaptic weighting in the striatum (in reinforcement learning models of the BG). In contrast, initial disinhibition in the LDDM is broad and non-selective, activated across all option subcircuits (e.g., via broadcasting projection of acetylcholine or serotonin) and serving to transform circuit function from normalized value coding to WTA selection. Subsequently, because disinhibitory neurons are driven by local excitatory (*R* neuron) input and interact in a multiplicative fashion, disinhibition becomes selective and biased, similar to CBG models. A more subtle difference between the models lies in the structure of mutual competition: like other cortical decision models (i.e. RNM), the LDDM utilities lateral inhibition to provide competition between option-selective neurons; lacking clear evidence for such lateral connections in the BG, CBG models achieve competition in a more complicated manner that typically involves direct pathway disinhibition coupled with broad inhibition via the indirect and/or hyper direct pathways (for example, involving the subthalamic nucleus). This novel aspect of the LDDM, with regard to CBG models, is now highlighted in the manuscript.

We have now added additional text to the Discussion section (pp. 36-37) describing known disinhibition circuits in the CBG, discussing the link between the LDDM and previous computational models, and touching on the novel contribution of the LDDM:

“While largely absent in standard existing cortical decision models, disinhibition is a key element of action selection in models of the cortical-basal ganglia (CBG) system (Bogacz and Gurney, 2007; Frank, 2005; Lo and Wang, 2006; Schroll and Hamker, 2013; Wei et al., 2015). In the basal ganglia direct pathway, GABAergic neurons in the striatum inhibit neurons in the substantia nigra pars reticulata and internal globus pallidus, which in turn send inhibitory projections to the thalamus. Cortical inputs to the striatum thus produce a disinhibition of thalamic outputs to the cortex and brainstem motor areas, resulting in motor facilitation. Crucially, the activation of disinhibition in the CBG system is selective: the selection of a specific action requires a selective disinhibition driven by asymmetries in cortical inputs or striatal synaptic weights. This selective disinhibition is an essential element of computational models of the CBG system (Frank, 2005; Lo and Wang, 2006), including more complex models that incorporate global inhibition mediated by the indirect and hyper-direct pathways (Bogacz and Gurney, 2007; Schroll and Hamker, 2013; Wei et al., 2015).

While both the LDDM and standard CBG models utilize disinhibition to drive selection, they differ in two important ways. First, disinhibition in the LDDM specifically functions to implement a transition between value coding and WTA selection states. This transition is mediated by a broad/non-selective activation of disinhibition across the decision circuit. The activation of disinhibition is not biased towards specific alternatives until a period of interaction with differential value inputs to option-specific subcircuits that instantiates the WTA process. Second, disinhibition in the LDDM is tightly integrated with the lateral inhibition that mediates competition (and hence normalization) between alternatives; consistent with the microarchitecture of the cortex which it seeks to model (Fu et al., 2014; Karnani et al., 2016; Kepecs and Fishell, 2014; π et al., 2013; S. Zhang et al., 2014), disinhibitory, inhibitory and excitatory neurons are part of the same local circuit. In contrast, the basal ganglia are known to lack these local, lateral connections and mutual competition. As a result CBG models typically require both direct pathway disinhibition along with diffusive suppression of competing motor plans via the indirect or hyper-direct pathways (Bogacz and Gurney, 2007; Schroll and Hamker, 2013; Wei et al., 2015) for effective operation. Thus, while conceptually similar to the CBG models, disinhibition in the LDDM is in some ways quite distinct, tightly integrated with competitive inhibition, and providing dynamic control of circuit state, both characteristics of decision-making in cortical brain areas.”

3. I would recommend finding a non-DNM and non-RNM control model to compare the LDDM against.

We appreciate the suggestion to compare the performance of the LDDM against another model, particularly in terms of WTA activity. First, we wish to clarify a point of interpretation that we did not express clearly in the original manuscript. The RNM is not strictly speaking a component of the LDDM: a key feature of RNM models is they implement a pooled inhibition, in which different option-specific excitatory neurons receive the identical inhibition signal; in contrast, the LDDM has a more complicated inhibitory structure, since the disinhibition – once activated – is driven by local excitatory neuron activity and thus functionally segregates the inhibitory pools. At a practical level, RNM models predict non-selective inhibition that does not encode any decision-relevant information; in contrast, the LDDM predicts selective inhibition where the activity of a given *G* neuron will reflect choice-related information relevant to its associated *R* neuron. Importantly, recent empirical evidence suggests that inhibition in decision circuits is selective and structured rather than non-selective. Our revision makes this issue much clearer and focused on this novel feature of the LDDM.

However, we do appreciate that including an additional model for comparison is informative and helpful and so have added a fourth model to the paper. While RNM models are by far the most prevalent model of decision-making circuits, another well-known model is the leaky competing accumulator (LCA) model (originally proposed by Usher and McClelland, 2001). The LCA is an appropriate choice for model comparison here because it is: (1) a biologically-inspired model of choice, (2) specifically designed to capture choice data in perceptual decision tasks (such as used in the Roitman and Shadlen dataset), and (3) intended to capture both psychometric (choice) and chronometric (RT) aspects of relevant decision data.

The LCA model is now described in the Methods section (pp. 51-52):

“Another widely acknowledged decision circuit model – the leaky competing accumulator model (LCA) (Usher and McClelland, 2001) was fit to the behavioral data (Roitman and Shadlen, 2002). The dynamics of the two nodes in the LCA can be described using the following differential equations (Eqation 22).

where xi (i=1 and 2) indicates the activity of each node; ρi indicates the excitatory input value to each node; k indicates the net leakage on each node after the cancellation of recurrent excitation; β weighs the mutual inhibition strength from the other nodes; ξi is a Gaussian random noise on each node with a standard deviation of σ.

The input values ρ_i_ were set as 1+c’ for Option 1 and 1-c’ for Option 2, with c’ changing over 0 to.512. We fitted the threshold as a free parameter. In that way, the time constant τ can be taken as an arbitrary value (100 ms used in our case) since it was not independent from the threshold. Other than the parameters we mentioned above, non-decision time *T*_0_ was fixed as 120 ms, sharing the same assumption with the other two models based on the empirical observed delays after stimulus onset (90 ms) and before saccade (30 ms). That gives in a total of four free parameters to estimate *k*, *β*, *σ*, and *threshold*. Since the scale of the activities is arbitrarily defined, it would need rescaling when compared to the empirical data of mean firing rates in the unit of Hz. The task setting and the optimization used were kept the same as in fitting the LDDM (see above). The time step *dt* was set as .001 s.”

Results of the model comparison between the LDDM, RNM, and LCA models are now described in the Results, and the fitted LCA data are shown in a new Figure 6—figure supplement 5. The following text has been added to the Results section (p. 20, lines 424-432):

“We compared the performance of the LDDM in fitting this classical dataset with the reduced form of the RNM (Wong and Wang, 2006) (Figure 6—figure supplement 4), as well as another prominent computational decision model with a similar architecture of mutual inhibition – the leaky competing accumulator model (LCA; Usher and McClelland, 2001; see Figure 6—figure supplement 5). The performances of the three models were close in predicting averaged RTs and choice accuracy (panel **C**). However, the LDDM captures the skewness and the shape of RT distributions better than the other two, as reflected in goodness of fit (negative log-likelihood) and AIC measures (nLL_LDDM_ = 16546, nLL_RNM_ = 16573, nLL_LCA_ = 16948, AIC_LDDM_ = 33109, AIC_RNM_ = 33165, AIC_LCA_ = 33932).”

4. I recommend using model fit metrics to evaluate how well the LDDM (and a control model) explain the neurophysiological data.

We appreciate the Reviewer’s constructive recommendation. We now clarify that we examined model *neural* activity in three different models (LDDM, RNM, and now LCA) best fit to behavioral data. Adequately fitting the model (s) to neurophysiological data – particularly the fully dynamics of decision-related activity – will require more work and empirical observations, and given current technical limitations we do not address fitting the models to neural data in the current manuscript.

However, we believe there are informative aspects of the approach we currently take – fitting the models to behavioral (choice and RT) data, and then examining the neural activity of the best fitting model units. We now quantify how well different model excitatory unit activity – with model parameters fit to behavior – matches empirically recorded data (from Roitman and Shadlen), focusing on how population average activity varies with stimulus level (coherence) at different timepoints in the trial. Because these models were not fit to the neural data, we use simple RMSE as a measure of explanatory power.

This information is now presented in Figure 6E, 6-S4E, and 6-S5E, along with new Results text (pp. 20-21, lines 443448 and lines 450-461):

“More quantitatively, we examined the relationship between activity and coherence at the specific time point reported in the original work (arrow points a and b, Figure 6E). Model predictions align well with empirical observations: across the three alternative models, the deviation between empirical recordings and model-predicted activity is the smallest for LDDM (quantified by root-mean-square error (RMSE); RMSE_LDDM_ = 2.74 (Figure 6E), RMSE_RNM_ = 20.10 (Figure 6—figure supplement 4E), RMSE_LCA_ = 3.92 (Figure 6—figure supplement 5E)).

Aligned to the onset of decision (Figure 6D, right), … Quantification shows that LDDM again best predicted empirical neural activity with data aligned to choice onset (RMSE_LDDM_ = 6.77 (Figure 6E); RMSE_RNM_ = 9.35 (Figure 6—figure supplement 4E); RMSE_LCA_ = 7.51 (Figure 6—figure supplement 5E)). Thus, *R* unit activity – in a model with parameters fit only to behavior – replicates the recorded activity of parietal neurons during both initial decision processing and eventual choice selection.”

On a broader note, a significant limitation of fitting neural data and using model comparison metrics is that the novel focus of the model – the activity of inhibitory and disinhibitory neurons – has not been empirically well quantified in these types of tasks. Vary little data from these classes of neurons is currently available. In the revised manuscript, we address this limitation by describing model predictions about the dynamics of different neuronal types and the perturbation outcomes to guide future data collection. We hope that such model predictions, along with future work recording from identified neural subpopulations, will help deepen the understanding of circuit mechanisms underlying the decision-making process.

To highlight this issue we now visualize the model predictions regarding the dynamics of the inhibitory units (*G*) and disinhibitory units (*D*) for future empirical testing (please see revised Figure 6F-I). We have also included a description of the informative pattern in the predicted dynamics of *G* and *D* in Results (pp. 21-22).

“Unlike the RNM and LCA models, the LDDM predicts different dynamics in different subtypes of interneurons (Figure 6F-I). The inhibitory (*G*) units selectively code input values and choice but exhibit complex dynamics due to the interplay of feedforward excitation, lateral inputs, and disinhibition. Early on (dynamics sorted to the left in Figure 6F and upper panel in Figure 6G), the *G* activities initially increase due to excitatory drive from *R* units. Later on, when the inhibition from *D* units increases (Figure 6H), the *G* activities start to decrease. Near the time of choice (dynamics sorted to the right in Figure 6F and the lower panel in Figure 6G), the chosen *G* units show lower activities than the unchosen side because of stronger inhibition from *D* as an outcome of WTA competition. The dynamics of *D* units rapidly increase in the early stage, driven by excitatory *R* unit activity (dynamics sorted to the left in Figure 6H). Dynamics in the late stage (dynamics sorted to the right in Figure 6H) show higher activity on the chosen side than the unchosen side as an outcome of WTA competition. Both types of interneurons show different time-dependent patterns of coherence-dependence that likely reflect the complex dynamics of the system and RT-based data aggregation methods (Figure 6G, H). While the activity of different interneuron subtypes have not been widely recorded in decision tasks, these new LDDM predictions provide a testbed for future empirical and theoretical investigations.”

Reviewer #3 (Recommendations for the authors):1) I think the authors will have to rewrite parts of the manuscript to address the concerns I raise – (i) especially to clarify the precise interpretation of the single key parameter that determines the behavior of the model, and (ii) point out the connection to previous work (Machens et al. 2005, Yang et al., 2016, Litwin-Kumar et al. 2016) emphasizing the specific ways in which their work is an advance on these previous studies.

We thank the Reviewer for these suggestions, and have significantly rewritten parts of the manuscript to address these concerns.

One major point was the conceptual interpretation of the β parameter, which in the LDDM controls the degree of disinhibitory drive in the network and consequently governs the shift between value representation and WTA selection.

The Reviewer rightly points out connections to past literature, both in terms of the Machens et al. mutual inhibition model and its flexible dynamic states and other existing models with disinhibitory motifs that share some functions with the LDDM. Regarding the Machens model, it is an important related model: like the LDDM, it models a decision process with mutual inhibition as the key competitive mechanism, and it can flexibly reconfigure its dynamics to transition from stimulus encoding (point attractor) to working memory (line attractor) to WTA selection (saddle point). We do note that there are some key differences between the LDDM and the Machens model: (1) it models a sequential two-interval decision rather than a simultaneous decision, (2) while it achieves a reconfiguration of state via changes in external input, as in the LDDM’s top down activation of disinhibition, it requires a distinct switch in functional connectivity between inputs and decision circuit elements, and (3) disinhibition plays an ancillary rather than central role (it is presented as a possible input-switching mechanism). We have added new text about the Machens et al. model in the Discussion section (p. 38, lines 826-837):

“The LDDM achieves the flexible reconfiguration of line attractor and point attractor states under the control of disinhibition, suggesting that attractor dynamics might not be a fixed property of a network; rather, it may be adaptive and controllable by a top-down signal operating via gated disinhibition. Of course, similar reconfiguration has been achieved by other important circuit mechanisms that have been well-described. For example, a mutual inhibition network can capture the different regimes of sequential two-interval decision-making – stimulus loading, working memory, and comparison – by assuming a flexible reconfiguration of the external inputs (Machens et al., 2005). Similar to the LDDM, this model can transition between point attractor (initial stimulus encoding), line attractor (working memory), and saddle point (comparison) dynamics. Interestingly, disinhibition may also play a role in this model, by providing a theoretical mechanism to switch the routing of external inputs within the circuit, which drives the switch from line attractor to comparison dynamics.”

As the Reviewer notes, there are existing models that show a link between disinhibition and functions integrated together in the LDDM (divisive normalization, working memory, and gating/selection). We agree that it is important to relate the LDDM to prior work, and have added new text to the Discussion that discusses the relevant models raised by the reviewer (Litwin-Kumar et al., Kim and Sejnowski, Yang et al) as well as other predecessor literature (pp. 35-36, lines 761-779):

“The contribution of LDDM relative to existing disinhibition models

Disinhibition has been previously linked in separate models to several of the computational functions that are exhibited in a unified manner by the LDDM. For example, a computational model employing dendritic disinhibition captures flexible information routing in a context-dependent decision task, with dendritic disinhibition gating on specific inputs to a circuit while gating off other pathways (Yang et al., 2016). However, disinhibition plays a different role in this model (context-dependent input gating) from that employed in the LDDM (transition from value coding to WTA selection and mutual competition). In another example, PV neuron activation within a disinhibitory circuit motif can produce a divisive normalization of tuning curves in a model of visual cortex (Litwin-Kumar et al., 2016). This specific model of division, however, arises from different circuit mechanisms than those we employ, such as reduced tuned input and firing rate nonlinearities. Finally, disinhibition has also been proposed to underlie the long timescales of information processing seen in working memory, as enhancing inhibitory-to-inhibitory connections stabilize temporal dynamics and improve working memory performance in recurrent neural networks (R. Kim and Sejnowski, 2021). One other notable difference between previous research and our current work is that disinhibition in past models typically contributes to a specific function (e.g., input gating, categorical selection, working memory, etc.), whereas disinhibition in the LDDM both mediates a transition from value coding to WTA selection and plays an integral role in the selection process itself. Taken together, previous results and our current work reinforce the importance of incorporating disinhibition in circuit models of decision-making.”

2) It would also greatly help if the usage of notation is made consistent throughout the paper. For instance, in the figures and the equations in the main text (Equations 1-3), disinhibition is denoted as D, but in the methods (Equations 5-8) and the supplementary figures (Figure 2, Supplementary Figure 1) it is denoted as 'I'.

We thank the Reviewer for pointing this issue out. This confusion arose in the original manuscript because the equations in the Methods referred to a more general expanded version of the model, from which we derived the circuit model presented in the paper (LDDM). We have corrected the Equations 5-8 to become Equations 28-32, to convey our idea more clearly. In the expanded models, we added *D* units to the testing motifs, separated from the *I* units in the old version. *D* units represent disinhibition modules for local disinhibition and *I* units represent inhibitory modules for cross inhibition. In the revised version, the meaning of *D* units in the expanded models is now consistent with the meaning of *D* units in the LDDM, throughout the Results and the Methods sections. Please see the revised Methods (pp. 60-62).

3) I appreciate that the authors also studied a more general and 'extended' version of their model (of which the LDDM is a special case) and explore how it behaves in different regions of parameter space (Figure 2, Supplementary Figure 1). However, I found the general description of their extended model quite confusing, particularly, some of the design choices. For instance, the extended model consists of additional excitatory units (E) that are referred to as 'gain control boost loops'. These are never mentioned in the main text and their purpose for the overall story of the paper seems somewhat unclear to me. Since the R units already have projections to both 'local' and 'lateral' gain units (through 'omega', Figure 2A), couldn't the E units simply be replaced by stronger self-recurrence on the R units?

We apologize for any confusion in the original version of the manuscript, which likely arose because the previous version overemphasized the role of the extended model. Our paper focuses on the LDDM: a simple lateral inhibition model with recurrent excitation and an activatable within-option disinhibition. However, to emphasize that this model architecture was not selected in an arbitrary fashion, we included a brief synopsis of how we narrowed down a broader range of possible models to the version characterized in the rest of the paper (LDDM), based on desired functional characteristics (e.g., WTA activity).

The presentation of the extended version of the model is intended to show all possible modifications that were tested, and eliminated, before we settled on the LDDM architecture. In the main text, we do not discuss other motifs in the service of the readability of the paper. We realize that there were also some problems with notation inconsistency between the original Methods section and the original Results section. We have now corrected these notation problems, as we mentioned in the above point.

Regarding the E units, the reviewer is correct: adding an *E* unit to project selectively to the lateral *G* units will be very similar to selectively changing the connection weights matrix ω_ij_. However, it will be different from simply changing the self-excitation on the *R* units (i.e., α), since each *R* unit receives gain control from lateral *R* units (mediated via *G* units) but also receives gain control from itself (via the local projection to *G*). As an outcome, changing the strength of self-excitation is not alone sufficient to break the balance between the two *R* units, i.e., will not lead to winner-take-all competition. Only when the ω_ij_ matrix is asymmetric – equivalent to introducing a new *E* unit to selectively target the lateral *G* – will the circuit be able to break the balance of self gain control and lead to winner-take-all competition.

We have now clarified these issues in the revised Methods (pp. 60-62):

“Motifs tested and compared for normalized coding and winner-take-all choice

We tested a series of motifs and found local disinhibition is critical for the integration of normalized valuation and choice functions. To do this, we tested four types of modifications that might enhance mutual competition between the option-specific local sub-circuits (Figure 2—figure supplement 1A): (a) *Recurrent self-excitation* (loops weighted by α), with self-amplification of each *R* unit, a property shown to be important for mutual competition in the RNM. (b) *Local disinhibition* (loops weighted by β), which is the focus of the main text, mediated through disinhibitory units (*D*); the function of a *D* unit is to inhibit the gain control *G* unit in the local sub-circuit therefore release inhibition on the local *R* units. (c) *Cross inhibition* (loops weighted by η), which directly inhibits the lateral *R* units through inhibitory units (*I*) to implement mutual inhibition. (d) *Lateral gain control boost* (loops weighted by γ), which is mediated through excitatory units (*E*) to boosts the lateral *G*, therefore drives higher gain control on the lateral *R* than the local *R* (i.e., asymmetric gain control) and realizes mutual inhibition.

[…]

The active and inactive states of the four types of loops can be combined into 2^4^ = 16 possible models. Example dynamics were shown in Figure 2—figure supplement 1B for each type of model. When local disinhibition (β) is off (left two columns), the model generates WTA dynamics only when cross inhibition (η) is on. But the maximum activity in the late stage is still restricted to a value lower than the phasic peak during the early stage, contradicting empirical findings that the late stage decision threshold is usually higher than activity in the early phasic peak (Churchland et al., 2008; Kiani et al., 2008; Kiani and Shadlen, 2009; Louie et al., 2011; Roitman and Shadlen, 2002; Rorie et al., 2010; Shadlen and Newsome, 2001; Sugrue et al., 2004). This restriction arises because, with only cross inhibition, local option gain control is not released; this release requires local disinhibition. With local disinhibition on (β > 0, the right two columns), the models generate WTA dynamics with high activity in the late stage to reach the decision threshold. This is robust even without any other modifications (see the panel with η and γ off), highlighting the role of local disinhibition in generating WTA competition. For the sake of simplicity, we omitted other non-essential modifications and kept only the loop of local disinhibition. Because recurrent excitation is important for persistent activity and exists widely in cortical circuits, we retained it as well. The modified DNM model with local disinhibition and recurrent self-excitation is the primary model (LDDM) characterized in the current work.”

4) The most interesting analyses in the paper are where the authors fit the circuit model to neurophysiological data. The authors then report the values of the fitted parameters and also perform the model comparisons by reporting AIC/likelihood ratios. However, if possible, it would be very informative to also visualize the optimization surface of these fits to understand whether some of the free parameters trade-offs against one another, as I think that would affect the overall conclusions drawn in the paper, and also convince me about the robustness of the fitted parameters.

We agree with the Reviewer that examining likelihood surfaces is informative for understanding the relationship between parameters in fitting the dataset at hand (note that we fit the model to the behavioral data, with neurophysiological activity derived from the behaviorally-fit model). In the revision, we now visualize the loglikelihood in the spaces of different pairs of parameters as shown in new figure supplements (Figure 6—figure supplement 1). All of the spaces show smooth and single maximum topography, which suggests the model fitting is robust and stable. All of the parameters show no extreme collinearity and identifiable maximum value. α and β show mild collinearity, Author response image 1. The best fitting parameters visualized in the grids precisely match the best-fitting parameters given the precision of the grids.

**Author response image 1. sa2fig1:** The shape of predicted reaction-time distribution over a wide range of α and β values by LDDM. Each grid indicates the predicted RT histogram normalized in the range of minimum and maximum RTs. The shape of RT distribution exhibits a pattern of increasing skewness when α increases and decreasing skewness when β increases.

5) A somewhat more open-ended question is about the choice of the time constants for the 3 types of units in the model (R, G, and D), which appear to be fixed to a value of 100ms for all the results presented in the manuscript. Can the authors justify this choice? Considering that SST (which, I presume are the gain control units G) and VIP neurons have fundamentally different conductance profiles and are known to show an entire range of spiking patterns (Tremblay et al., 2016), is it justifiable to assume that their time constants all have the same value?

We agreed with the Reviewer that the time constants for different units could (or should) possibly be different, given the biological assumptions of different neuronal types. Thus, we fitted the τ of each unit as free parameters in the model fitting to Roitman and Shadlen’s data. We used fixed τ (100 ms) only for visualizing example dynamics when they are not quantitative results (e.g., the example dynamics shown in Figure 2). In addition, in the equilibrium analyses, time constants do not affect the equilibria of the circuit and thus the exact values we assumed on the τ does not impact our conclusions. We have clarified this issue in our revised Results (p. 19, lines 402-407) and Methods (p. 44, lines 945-946).

“The model is then reduced to seven parameters: recurrent excitation weight α, local disinhibition weight β, noise parameter σ, input value scaling parameter *S*, and time constants τ*_R_*, τ*_G_*, and τ*_D_* (see Methods for model-fitting details). Predictions of the best fitting model are shown in Figure 6A (best fitting parameters: α = 0, β = 1.434, σ = 25.36, *S* = 3251, τ_*R*_ = .1853, τ_*G*_ = .2244, and τ_*D*_ = .3231).”

“τ*_R_*, τ*_G_*, and τ*_D_* were set as the same value of 100 ms only for non-quantitative visualization purposes and fitted independently as free parameters in the model fittings.”

6) In general, the presentation of the figures can be improved:a) In Figure 2—figure supplement 1, I should be replaced by D. Also, the parameter γ seems to be missing from the rows in subpanel B of this figure.

At the Reviewer’s suggestion, we have improved these illustrations and replaced the missing information. Please see the revised Figure 2—figure supplement 1. Thanks for pointing this out.

b) In Figure 5, it's hard to follow which subpanels are the 'main' subpanels and which ones are the insets.

We have revised Figure 5 on the sake of clarity, removing the inset subpanels since these are known results/properties from the RNM and have been reported in the cited literature.

c) The legend of Figure 4B (right column) seems to have an extra set of dots (ones that indicate the legend of V_in)

We have corrected the mistake in the illustration, thank you.